# Blood stem cell-forming haemogenic endothelium in zebrafish derives from arterial endothelium

Florian Bonkhofer[1], Rossella Rispoli[1,2,5], Philip Pinheiro[1,5], Monika Krecsmarik[1,3], Janina Schneider-Swales[1], Ingrid Ho Ching Tsang[1], Marella de Bruijn[1], Rui Monteiro [1,3,4], Tessa Peterkin[1] & Roger Patient[1,3]

Haematopoietic stem cells are generated from the haemogenic endothelium (HE) located in the floor of the dorsal aorta (DA). Despite being integral to arteries, it is controversial whether HE and arterial endothelium share a common lineage. Here, we present a transgenic zebrafish *runx1* reporter line to isolate HE and aortic roof endothelium (ARE)s, excluding non-aortic endothelium. Transcriptomic analysis of these populations identifies Runx1-regulated genes and shows that HE initially expresses arterial markers at similar levels to ARE. Furthermore, *runx1* expression depends on prior arterial programming by the Notch ligand *dll4*. Runx1$^{-/-}$ mutants fail to downregulate arterial genes in the HE, which remains integrated within the DA, suggesting that Runx1 represses the pre-existing arterial programme in HE to allow progression towards the haematopoietic fate. These findings strongly suggest that, in zebrafish, aortic endothelium is a precursor to HE, with potential implications for pluripotent stem cell differentiation protocols for the generation of transplantable HSCs.

[1] Molecular Haematology Unit, Weatherall Institute of Molecular Medicine, John Radcliffe Hospital, University of Oxford, Oxford OX3 9DS, UK. [2] Division of Genetics and Molecular Medicine, NIHR Biomedical Research Centre, Guy's and St Thomas' NHS Foundation Trust and King's College London, London, UK. [3] BHF Centre of Research Excellence, Oxford, UK. [4] Institute of Cancer and Genomic Sciences, University of Birmingham, Birmingham B15 2TT, UK. [5] These authors contributed equally: Rossella Rispoli, Philip Pinheiro. Correspondence and requests for materials should be addressed to R.M. (email: r.monteiro@bham.ac.uk) or to R.P. (email: roger.patient@imm.ox.ac.uk)

Definitive haematopoietic stem cells (HSCs) drive lifelong reconstitution of the blood system and are pivotal in the treatment of haematological disorders including leukaemia[1]. During embryogenesis, HSCs are generated from an endothelial precursor termed haemogenic endothelium (HE)[2–5]. Replication of this process will facilitate the generation of transplantable HSCs in vitro from pluripotent stem cells (PSCs). However, despite recent advances in generating blood progenitors with broad potential[6], the in vitro generation of transplantable HSCs without genetic manipulation has yet to be reported[7,8]. This likely reflects the complexity of embryonic haematopoiesis occurring in multiple tissues with differing genetic programmes[1,9]. The mouse embryo harbours at least two types of HE, in the extra-embryonic yolk sac (YS) and the embryo proper[10]. HE with the potential to give rise to HSCs is found mainly in the floor of the dorsal aorta (DA)[11], but whether HE derives from arterial endothelium or represents an independent lineage is controversial[12–14].

The transcription factor Runx1 is expressed in HE and is essential for the emergence of HSCs[15,16]. Runx1 is required for the endothelial to haematopoietic transition (EHT)[17,18], leading to the formation of intra-aortic clusters (IAC) containing HSCs[19]. Mechanistically, Runx1 facilitates EHT by repressing the endothelial programme through upregulation of the transcriptional repressors, Gfi1/Gfi1b[18]. However, Runx1 was shown to still be required for HSC formation even after most HE cells have differentiated into IAC cells, indicating that RUNX1 regulates as yet unknown, critical target genes[20].

Here, we present a detailed in vivo analysis of the aortic HE based on a newly generated zebrafish runx1 reporter line. In addition to the identification of HE-specific Runx1-activated genes, we show that arterial gene expression in the HE is negatively regulated by Runx1. HE not only maintains an arterial programme in runx1$^{-/-}$ mutants, but it is also dependent on prior arterial specification, strongly suggesting that HE derives from aortic endothelium.

## Results

**A TgBAC(runx1P2:Citrine) reporter labels aortic HE.** To characterize the aortic HE and to screen for Runx1 targets, we utilized the zebrafish model. Here, the runx1 gene structure is largely conserved, including regulation by two alternative promoters, P1 and P2 (Fig. 1a). However, in zebrafish, exon 2 is split into 2a and 2b by a non-conserved intron, positioned where mice/humans harbour an in-exon splice site (Fig. 1a and Supplementary Fig. 1a). Isoform-specific in situ hybridization (ISH) shows that runx1 expression in the HE is solely initiated through P2 activity (Fig. 1b, c). To ensure proper regulation in runx1 reporter fish, we made use of bacterial artificial chromosome (BAC) technology. Of the four previously identified zebrafish BACs containing runx1[21], we chose the 97a02 BAC which harbours both promoters plus substantial upstream sequence (Supplementary Fig. 1b). A Citrine cassette followed by a polyA stop signal was recombineered downstream of the P2 ATG (Fig. 1d) to establish a stable TgBAC(runx1P2:Citrine) line, which expresses Citrine under the control of runx1 regulatory elements.

During early haematopoietic development, Citrine expression recapitulated endogenous runx1 in the posterior lateral mesoderm (PLM) which contains precursors to endothelium and primitive blood cells, but lacked expression in runx1+ Rohon-Beard neurons (RBN) (Fig. 1e). Notably, Citrine transcripts were detectable at least 1–2 h before endogenous runx1 (Supplementary Fig. 1c), likely caused by a shorter transcription cycle. This allowed time for strong Citrine fluorescence to be detected in the migrating PLM from around 12–13 hpf (Supplementary Fig. 1d), coincidental with the detection of runx1 expression by ISH.

During the subsequent definitive haematopoiesis, runx1 is first detectable in the aortic HE by ISH from ~23/24 hpf onwards[22]. Importantly, Citrine RNA was detectable in the HE at 24 hpf (Fig. 1f), Citrine fluorescence mimicked expression of runx1 (Fig. 1g, h) and no difference in runx1 expression was detectable between reporter and wild-type embryos (Supplementary Fig. 1e). Unlike the earlier Rohon-Beard neurons, Runx1+ spinal cord neurons also expressed Citrine, suggesting that some neuronal enhancers are included in the BAC construct. Around 50 hpf, when most HE cells had initiated EHT[2], Citrine+ cells were found in the sub-aortic space (Supplementary Fig. 1f) and in the caudal haematopoietic tissue (CHT) (Fig. 1i), with the kdrl:mCherry transgene delineating the endothelial domain. Over subsequent days, Citrine+ cells were detectable within all haematopoietic niches including the CHT, the thymus and the kidney (Supplementary Fig. 1g). Morpholino (MO) knock-down of the upstream regulators, tal1b and gata2b[23,24], further validated proper regulation of the BAC-driven runx1P2:Citrine transgene for haematopoietic tissues (Supplementary Fig. 1h). Overall, we present a runx1 reporter line suitable for the analysis of aortic HE in vivo.

**Isolation of enriched haemogenic and aortic roof endothelium.** To analyse the DA region in more detail we used TgBAC (runx1P2:Citrine);Tg(kdrl:mCherry) double-transgenic embryos (Fig. 2a) at 29 hpf. Co-presence of mCherry in Citrine+ cells of the DA lining confirmed their endothelial identity (Fig. 2b). Unexpectedly, we detected additional weak Citrine fluorescence in the aortic roof endothelium (ARE) and sprouting inter-somitic vessels (ISV), but not in non-aortic endothelium (NAE) including the cardinal vein (CV) and veins and arteries in the head and tail. Immunohistochemistry confirmed the presence of stable Citrine protein in the ARE, whereas RNA expression was only detectable in the DA floor (Supplementary Fig. 2a). The weak Citrine fluorescence in ARE diminished over the next ~24 hours (Supplementary Fig. 2b) indicating high protein stability, also seen in primitive erythrocytes (eryP) (Supplementary Fig. 2c). These findings indicate a carryover of residual Citrine protein to the ARE from earlier stages of DA development and suggest a lineage relationship between runx1+ and runx1− aortic endothelium.

Next, we analysed Citrine transgene expression during DA lumenisation at 18–22 hpf. Fluorescent ISH (FISH) highlighted that runx1 is expressed at 22 hpf (Supplementary Fig. 2d) with the first positive cell detectable at 20 hpf. Importantly, we also detect Citrine RNA expression already at 18–20 hpf and Citrine fluorescence from ~20 hpf onwards in most DA cells, before expression became restricted to the DA-floor (Supplementary Fig. 2e–g), likely being the cause of the carryover of Citrine protein detected in the ARE (Fig. 2b). We propose that DA angioblasts, upon reaching the midline and just before lumenisation, experience the first runx1 inductive signals, including BMP[25], detected by our runx1P2:Citrine reporter. As a consequence of lumenisation, cells in the floor of the DA will stay proximal to the ventral BMP source and continue to upregulate the Citrine reporter, as well as endogenous runx1 to levels detectable by ISH, while cells in the roof are positioned distal to the BMP source and lose expression of Citrine and runx1.

To isolate the HE, here operationally defined as runx1+ endothelium of the DA-floor, a FACS gating strategy was established (Fig. 2c). runx1P2:Citrine+kdrl:mCherry+ double positive cells with high runx1P2:Citrine expression (DP-R1$^{hi}$) were discriminated from cells with lower Citrine intensity (DP-R1$^{med}$ and DP-R1$^{lo}$). Cells single-positive for kdrl:mCherry+ (SP-kdrl), representing a mixture of venous and arterial cells, served as a NAE control. Molecular characterisation at 29 hpf

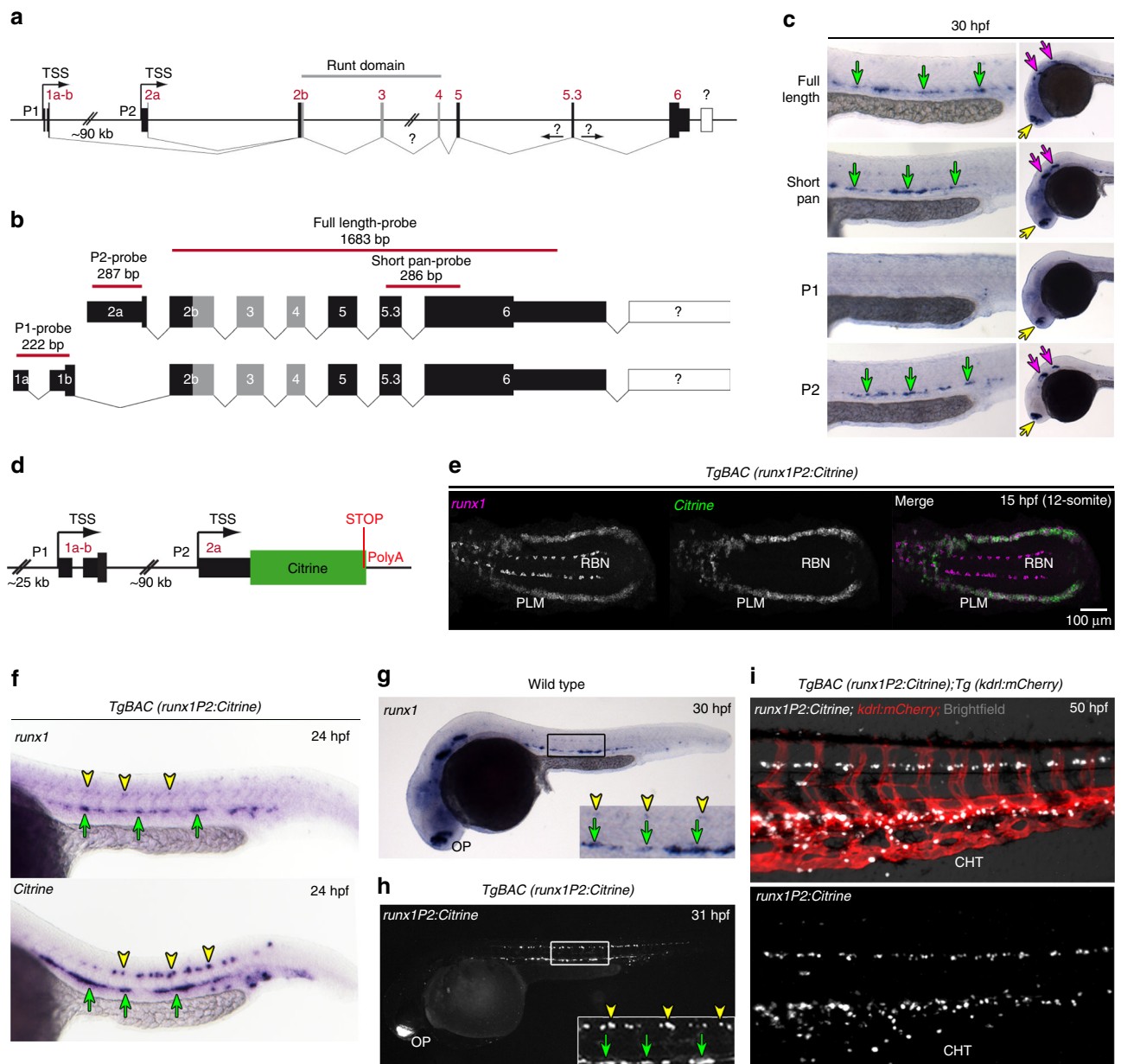

**Fig. 1** Generation of a zebrafish BAC transgenic reporter line for *runx1*. **a** Schematic representation of the zebrafish *runx1* locus. Exon nomenclature adjusted to the human *RUNX1* locus. P1 and P2 indicate the distal and the proximal promoter respectively (TSS: transcriptional start site). **b** Schematics of the two alternative transcripts derived from the alternative promoters P1 and P2. Transcript-specific and pan binding ISH probes are indicated with their respective lengths. **c** ISH for *runx1* isoforms in 30 hpf embryos. Left panel: trunk region. Right panel: head region. Green arrows point to the HE, yellow arrows show the olfactory placode and purple arrows depict neurons in the brain region. **d** Schematic of the recombineered 97a02 BAC. A *Citrine* reporter cassette was placed downstream of the P2 ATG. **e** Confocal microscopy image of a flat mounted 15 hpf embryo after double FISH for *runx1* and *Citrine* in *TgBAC(runx1P2: Citrine)* depicting the region of the posterior lateral plate mesoderm (PLM) and Rohon-Beard neurons (RBN). Maximum intensity projection of a 58 µm stack. **f–h** Gene expression analysis for *runx1* and *runx1P2:Citrine* during definitive haematopoiesis. Green arrows point to the HE. Yellow arrowheads point to neurons in the spinal cord. **f** ISH for *runx1* or *Citrine* in 24 hpf *TgBAC(runx1P2:Citrine)* embryos. **g** ISH for *runx1* in 30 hpf embryos. **h** Representative fluorescent microscopy image of a 31 hpf *TgBAC(runx1P2:Citrine)* embryo. Insets in (**g** and **h**) enlarge the boxed region. **i** Fluorescent microscopy image of a 50 hpf double transgenic *TgBAC(runx1P2:Citrine);Tg(kdrl:mCherry)* embryo focusing on the region of the caudal haematopoietic tissue (CHT)

(Fig. 2d) detected high expression of *tal1* in DP-R1med and DP-R1hi cells, while expression of the HE marker genes *cmyb* and *gfi1aa* were exclusive to the DP-R1hi population. In contrast, expression of the DA-roof marker, *tbx20*, was depleted from DP-R1hi cells. Arterial marker genes *efnb2a* and *dll4* were gradually lost with increasing intensity of Citrine fluorescence, in agreement with a previously described early loss of endothelial potential in HE that coincided with *Runx1* expression[26]. Notably,

*dll4* showed highest expression in the DP-R1lo population, reflecting its arterial identity.

Quantification of flow-sorted populations detected 18.2 (+/− 3.0) DP-R1hi cells per embryo (Supplementary Fig. 3a–c), representing ~1/5 (21.2% +/− 2.7) of the whole DA (including ISV). These counts agree with previous estimates, including those showing that the DA circumference consists of no more than 4 cells[2] with ISV sprouts (2–4 cells) found every 2–3 aortic cells.

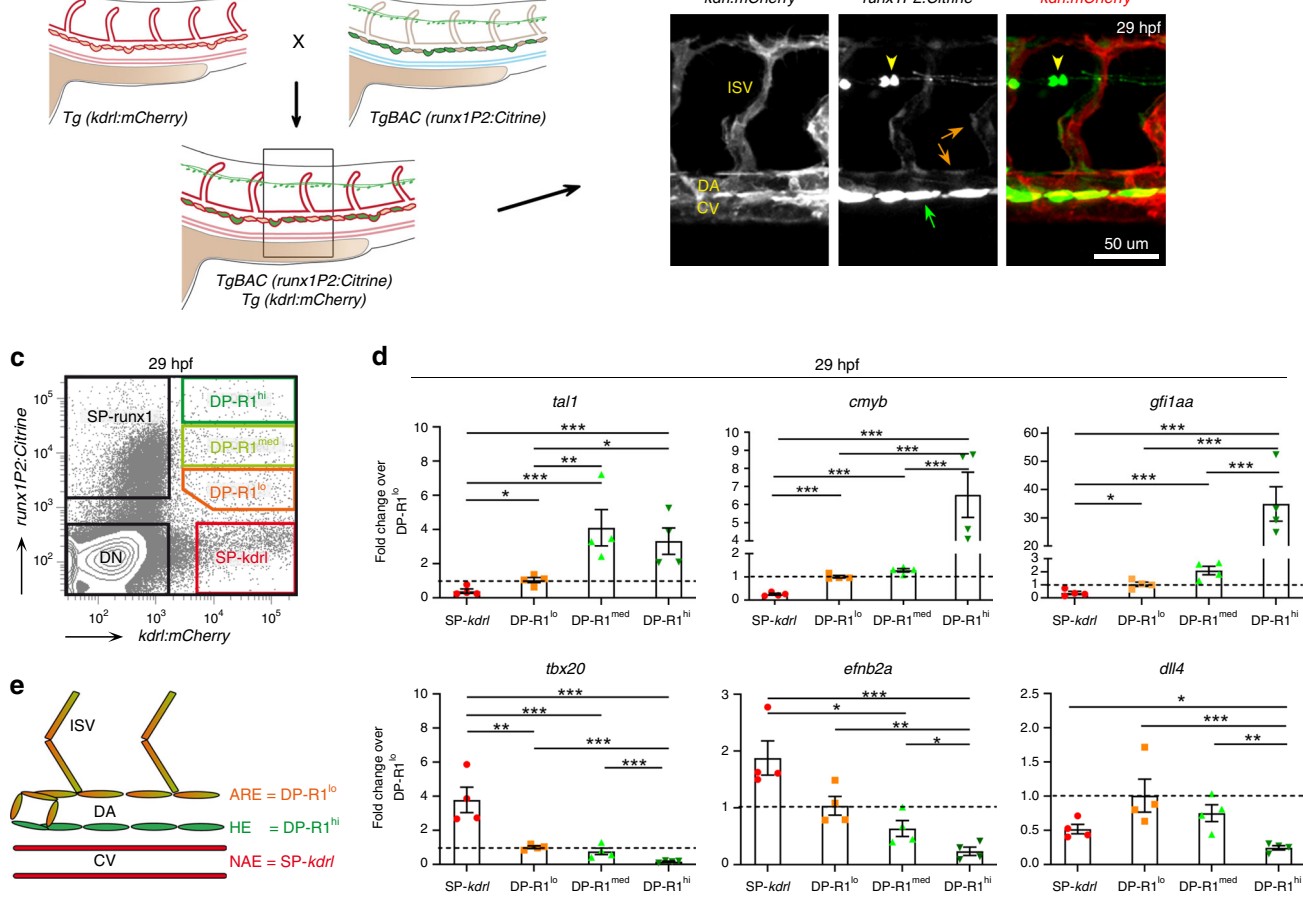

**Fig. 2** FACS based isolation of haemogenic and aortic roof endothelium. **a** Schematic of transgene expression in the trunk region of double transgenic *TgBAC(runx1P2:Citrine);Tg(kdrl:mCherry)* embryos during definitive haematopoiesis. **b** Representative confocal image of the DA region of double transgenic *TgBAC(runx1P2:Citrine);Tg(kdrl:mCherry)* embryos at 29 hpf. Laser intensities enhanced to detect Citrine fluorescence in the DA roof and sprouting inter-somitic vessels (ISV) (orange arrows) in addition to the HE (green arrow). Yellow arrow heads: neurons in the spinal cord. **c** Characteristic FACS plot of the established gating strategy for the isolation of endothelial sub-populations (*runx1hikdrl+* [DP-R1hi]; *runx1medkdrl+* [DP-R1med]; *runx1lokdrl+* [DP-R1lo]; *runx1−kdrl+* [SP-*kdrl*]) from double transgenic *TgBAC(runx1P2:Citrine);Tg(kdrl:mCherry)* embryos. **d** qRT-PCR analysis of haematopoietic (*tal1*, *cmyb* and *gfi1aa*) and endothelial (*tbx20*, *efnb2a* and *dll4*) marker gene expression in the different cell fractions isolated from ~29 hpf embryos following the FACS gating strategy depicted in (**c**). Graphs show the measured mean fold change relative to the expression detected in the DP-R1lo fraction. $n = 4$ independent experiments. Error bars represent the SEM. 1-way ANOVA; *$p < 0.05$; **$p < 0.01$; ***$p < 0.001$. **e** Schematic of the vessel structure in the zebrafish trunk region discriminating between aortic roof endothelium (ARE) including the ISV, the HE and the non-aortic endothelium (NAE) including but not restricted to endothelial cells from the cardinal vein (CV). The depicted colour code refers to the colours of the FACS gates presented in (**c**)

Considering that most if not all DA floor cells are haemogenic[2], the ratio we detected (1/5 DP-R1hi to all DP cells) reflects the reported DA composition.

Altogether, the combination of confocal microscopy, gene expression analysis and flow cytometric counting strongly suggests that DP-R1hi cells represent the aortic HE, while DP-R1lo cells represent the ARE (Fig. 2e). Overall, our strategy successfully discriminated between HE and ARE while simultaneously excluding NAE.

**Early repression of the endothelial programme in aortic HE.** For further characterisation of the defined endothelial populations we performed RNA-seq analysis at 29 hpf (Fig. 3a, b), ~5 h after *runx1* becomes detectable by ISH and shortly before EHT. The number of mapped reads of *runx1* and *Citrine* as well as of *kdrl* and *mCherry* were well correlated (Supplementary Fig. 4a). Transcripts per million (TPM) values for the haemogenic genes *tal1*, *cmyb* and *gfi1aa*, tracked the gene expression patterns as detected by qPCR (Supplementary Figs. 4b and 2d). Expression of

the arterial markers *dlc*, *vegfc*, *dll4* and *hey2* was enriched in SP-*kdrl*, DP-R1lo and DP-R1med populations and depleted in the DP-R1hi. As expected, *nr2f2* expression was only enriched in the SP-kdrl population (Supplementary Fig. 4c). The low TPM values for *myod1*, *pdgfra* and *cdh17* suggest only minimal contamination by cells of the myotome, sclerotome/neural crest or pronephros, respectively, mostly within the SP-*kdrl* population.

Multidimensional scaling (MDS) analysis demonstrated that biological replicates clustered together and placed the HE further from the NAE than from the ARE populations on the first dimension (Fig. 3c). We found 5213 differentially expressed genes (DEG) (log2-FC < +/− 0.7) across all populations (one-way ANOVA, FDR < 0.05) (Supplementary Data 1). Consensus clustering analysis (Supplementary Fig. 5a–c) further supported the endothelial character of DP-R1lo cells that shared a substantial gene set with endothelial cells of the SP-*kdrl* gate (Cluster 1, 2570 genes). DP-R1lo cells also possessed a unique programme (Cluster 4, 242 genes) containing the arterial markers *dll4* and *hey2*. In agreement with the MDS plot analysis, DP-R1hi cells shared a

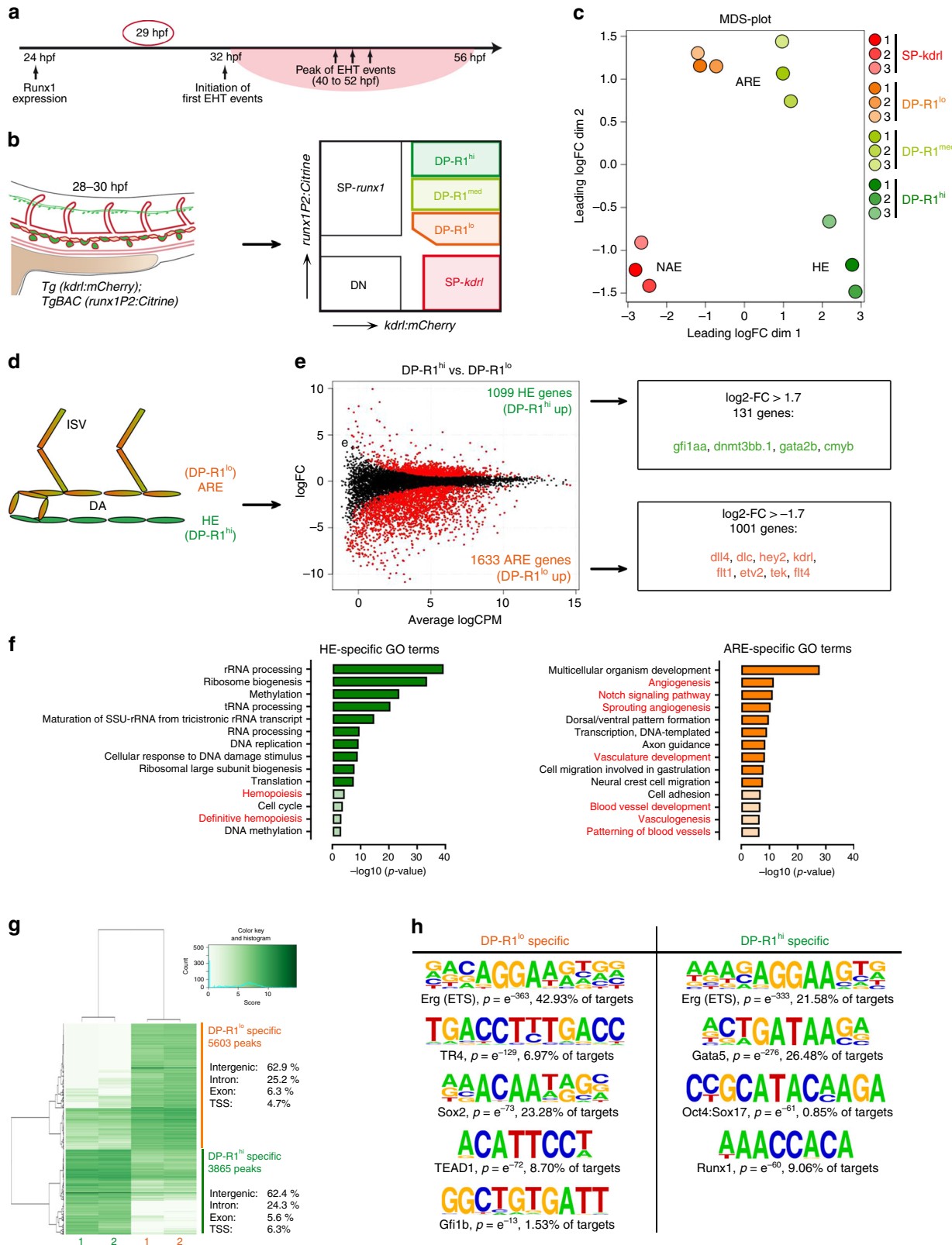

larger genetic programme with DP-R1$^{lo}$/DP-R1$^{med}$ cells (Cluster 2, 1305 genes) than with SP-*kdrl* cells (Cluster 5, 812 genes). In line with the detected carryover of Citrine protein to the ARE, these genetic commonalities hint at a close lineage relationship between the HE and ARE.

While most HE markers, *cmyb*, *gata2b* and *gfi1aa*[23,27,28] were part of Cluster 2, the previously identified HE-specific gene *dnmt3bb.1*[29] was not. To identify the most meaningful set of HE-specific genes we performed DEG analysis between the closely related HE (DP-R1$^{hi}$) and ARE (DP-R1$^{lo}$) (Fig. 3d, e), while

**Fig. 3** Characterization of genome wide gene expression in aortic HE. **a** Timeline depicting the most important events during definitive haematopoiesis in the zebrafish DA. At 29 hpf (red circle) cells of the HE have not yet initiated EHT but have been expressing *runx1* for several hours. **b** Experimental scheme to study HE and other endothelial sub-fractions by RNA-seq. Double transgenic *TgBAC(runx1P2:Citrine);Tg(kdrl:mCherry)* embryos were staged up to 28–30 hpf and taken forward for FACS based isolation of the indicated cell populations. RNA was isolated from at least 3,000 cells. $n = 3$ experiments have been performed. **c** Multidimensional scaling plot (MDS-plot) analysis of the cell populations depicted in (**b**). **d** Experimental strategy to identify HE-specific genes by comparing gene expression between HE (DP-R1$^{hi}$) and ARE (DP-R1$^{lo}$). **e** Left: Smear plot of the differentially expressed genes (DEGs) between DP-R1$^{hi}$ and DP-R1$^{lo}$. The number of significantly enriched genes for each population is indicated. Right: Number of significantly enriched genes with a log2-FC $> +/− 1.7$. Selected haematopoietic (green) and endothelial (orange) genes are shown. **f** GO term analysis on HE-specific (green) and ARE-specific (orange) gene-sets. Shown are the top 10 enriched terms (dark colour) plus additional informative terms (light colour). Haematopoietic and blood vessel development related terms are highlighted in red. **g** Heat map showing differentially open peaks between DP-R1$^{lo}$ and DP-R1$^{hi}$ cells as identified by ATAC-seq analysis. Genomic distribution of peaks is indicated. **h** de novo motif analysis underlying unique DP-R1$^{lo}$ or DP-R1$^{hi}$ specific peaks as indicated in (**g**)

excluding the mixed population of NAE (SP-*kdrl*). Overall, 2731 DEGs were identified (Supplementary data 2) of which 1099 genes were HE-specific (upregulated in DP-R1$^{hi}$). All known HE-marker genes were found within a subset of 131 genes with a log2-FC $> 1.7$ (Fig. 3e). HE-specific genes were enriched for gene ontology (GO) terms associated with cellular and metabolic processes (Fig. 3f), suggesting that the HE is in an active metabolic state, possibly adjusting its core machinery while transitioning from an endothelial to a haematopoietic identity. In contrast, within the 1633 ARE-specific genes (downregulated in DP-R1$^{hi}$) many endothelial and arterial genes were found (Fig. 3e). The ARE-specific genes were enriched for GO terms related to angiogenesis and vessel development (Fig. 3f), in turn indicating an early genome wide repression of the endothelial programme within the HE. Likewise, the top enriched KEGG pathway for ARE-specific genes was "Notch signalling pathway" (Supplementary Fig. 5d), agreeing with recent studies showing a repression of Notch signalling in the HE during the EHT and further haematopoietic development[30–32].

We further performed ATAC-seq analysis to study the open chromatin in DP-R1$^{lo}$ and DP-R1$^{hi}$ populations (Supplementary Fig. 5e, f). Overall, we identified 3865 peaks unique for the HE (DP-R1$^{hi}$) and 5603 peaks specific for the ARE (DP-R1$^{lo}$) (Fig. 3g; Supplementary Data 3). De novo motif analysis identified ETS binding motifs as the top enriched motive for both ARE and HE, reflecting their endothelial identity. ARE-specific peaks were also enriched for the Gfi1b binding motif, in agreement with the previously described Runx1-Gfi1 axis repressing endothelial genes[18] (Fig. 3h). Strikingly, HE-specific peaks were strongly enriched for the RUNX motif, implying a broad input for Runx1 required for HE progression to blood progenitors.

**Identification of HE-specific Runx1-regulated genes.** To identify Runx1-regulated genes, perturbation experiments were performed using a well-established *runx1* splice MO[2,22] that phenocopies the *runx1*$^{-/-}$ mutant[33] (Supplementary Fig. 6a,b). *TgBAC(runx1P2:Citrine)* morphants showed a loss of Citrine$^+$ cells in the CHT at 56 hpf. In contrast, the HE was widely unaffected at 35 hpf, in line with Runx1 being dispensable for establishing HE[18], thus allowing FACS-based isolation for subsequent RNA-seq analysis to identify Runx1 targets.

DEG analysis was performed between DP-R1$^{hi}$ populations of *runx1* MO and control conditions, as well as between the double-negative (DN) populations to correct for MO off-target effects (Fig. 4a). MDS plot analysis showed that DP-R1$^{hi}$ replicates clustered according to the presence/absence of the MO, while DN replicates all clustered together (Fig. 4b). Overall, we identified 676 DEG for the DP-R1$^{hi}$ population and only 176 DEG for DN cells (Supplementary Fig. 6c). Genes that correlated positively in both analyses were considered as off-targets and eliminated from the analysis (Supplementary Fig. 5d). Finally, we identified 245 Runx1-activated (MO-down) and 389 Runx1-repressed (MO-up)

genes (Supplementary data 4). Strikingly, we found the ARE phenotype to be repressed to a substantial extent by Runx1 (Fig. 4c).

To identify the most meaningful targets involved in definitive haematopoiesis we focused on 85 genes that were Runx1-activated and HE-specific (Fig. 4d). Of those, 22 genes showed evolutionarily conserved expression during the transition from murine HE to HSPCs[34,35] (Supplementary Table 1). Potential for direct Runx1 binding was shown for 16 of those genes by ChIP-seq in two haematopoietic cell lines, HCP7 and Kasumi-1[36–38]. We further analysed expression of the identified potential Runx1 targets in *runx1*$^{-/-}$ mutants[39] using the BioMark platform (Fig. 4e). We crossed the *kdrl:mCherry* and *runx1P2:Citrine* reporter lines onto a homozygous *runx1*$^{-/-}$ background and isolated the different subsets of the DA (Supplementary Fig. 7a). Gene expression was analysed at three time-points to additionally shed light on expression dynamics (Fig. 4f). For 11 of the 16 genes expression levels were shown to increase over time specifically in the HE and to be dependent on the presence of Runx1, also confirmed by ISH when functional probes were available (Fig. 4g; Supplementary Fig. 7b). These genes include five of the six zebrafish dnmt3 isoforms, including *dnmt3bb.1*, previously shown to be crucial for definitive haematopoiesis[29]. Other genes are the transcriptional repressor *gfi1ab*, the haematopoietic genes *syk*, *irf1b* and *pik3cd*, and the potential later niche genes *angpt1* and *mpl*, both expressed in adult HSCs and downregulated in *Runx1*$^{-/-}$ mutants[40–43].

Overall, the identification of HE-specific genes demonstrates that our experimental setup is a powerful tool to analyse gene expression dynamics within the HE and ARE. In combination with a *runx1*$^{-/-}$ mutant background we further identified several Runx1 targets for developmental haematopoiesis.

**Initial *dll4* expression in HE becomes repressed by Runx1.** Next, we used our experimental strategy dissecting HE from ARE to analyse the expression of additional factors involved in HSC development. Different ratios between Dll4- and Jag1-mediated Notch signalling define the arterial and haemogenic fates[44], whereby Jag1 is needed for HSC specification and a high Dll4 to Jag1 ratio drives the arterial programme[44–47]. We therefore analysed expression of *dll4* and *jag1a* in the zebrafish DA. In agreement with another study[47], highest expression of *jag1a* was detected in SP-*kdrl* cells (up to 4.2-fold higher compared to DP-R1$^{hi}$/DP-R1$^{lo}$), while a previously unidentified uniform expression was seen in the DA across the ARE and the HE (Fig. 5a). In contrast, *dll4* expression was dynamically regulated with an initial uniform expression in all fractions and an HE-specific down-regulation over time (Fig. 5b). In contrast, *runx1*$^{-/-}$ mutants expressed *dll4* at similar levels in the HE and the ARE at all analysed timepoints. ISH also showed a downregulation of *dll4* in the ventral DA from 27 hpf onwards (Supplementary Fig. 8a), but not in *runx1* morphants (Supplementary Fig. 8b) or *runx1*$^{-/-}$

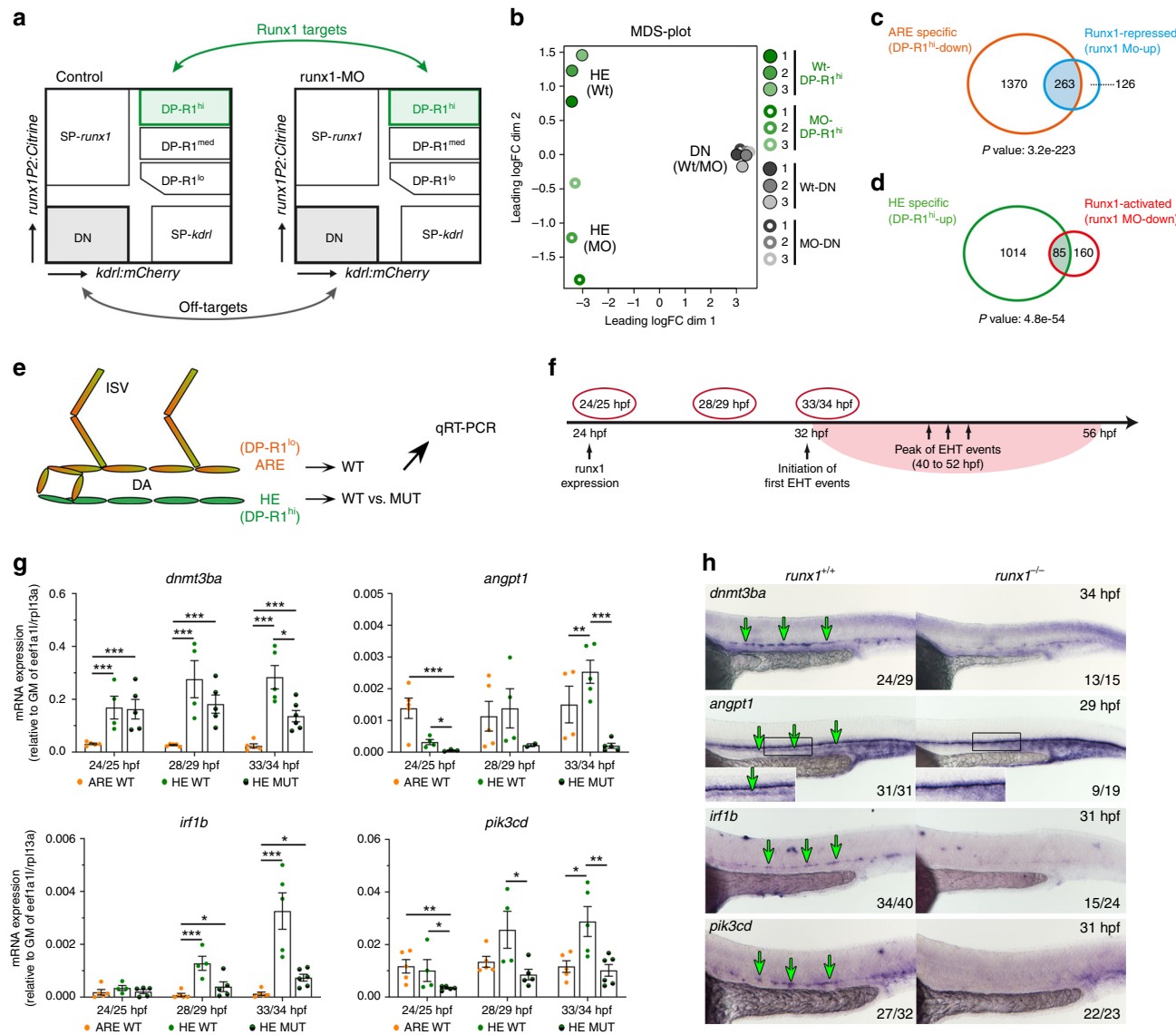

**Fig. 4** Identification of Runx1-regulated genes within the aortic HE. **a** Experimental scheme to identify Runx1-regulated genes in the HE (DP-R1hi). Highlighted cell populations (DP-R1hi and DN) from un-injected (control) and *runx1* MO-injected double transgenic *TgBAC(runx1P2:Citrine);Tg(kdrl:mCherry)* embryos (28–30 hpf) were used for differential gene expression analysis (DEG) by RNA-seq similar to Fig. 3. **b** Multidimensional scaling plot (MDS-plot) analysis of the cell populations depicted in (**a**). **c** Gene list intersection between ARE-specific and Runx1-repressed genes. **d** Gene list intersection between HE-specific and Runx1-activated genes. **e** Experimental strategy to analyse potential HE-specific Runx1 target genes by qRT-PCR in *runx1+/+* (WT) and *runx1−/−* mutant (MUT) embryos. **f** Timeline highlighting the time-points (red circles) used for subsequent qRT-PCR analysis. **g** qRT-PCR gene expression analysis of potential Runx1 targets (*dnmt3ba*, *angpt1*, *irf1b* and *pik3cd*) in the HE and ARE of *runx1+/+* (WT) and the HE of *runx1−/−* mutant (MUT) embryos at the time-points depicted in (**f**). Graphs show the mean of detected expression levels relative to the geometric mean (GM) of the two housekeeping genes *eef1a1l* and *rpl13a*. n = 5 independent biological experiments for WT embryos and n = 6 independent biological experiments for MUT embryos. Error bars represent the SEM. 2-way ANOVA; *$p < 0.05$; **$p < 0.01$; ***$p < 0.001$. **h** ISH analysis for the same genes analysed in (**g**) performed in *runx1+/+* and *runx1−/−* embryos

mutants (Fig. 5c). Importantly, double FISH showed co-localisation of *dll4* expression in regions of the HE also expressing high levels of *runx1* only in *runx1−/−* mutants but not in WT embryos (Fig. 5d). These findings suggest that zebrafish HE is derived from a *dll4+* progenitor and that *dll4* repression in the HE is Runx1-dependent.

Direct binding of Runx1 to the *Dll4* gene locus was found in previous ChIP-seq studies[36,38]. However, a likely candidate for an indirect regulatory link is Sox17, shown to directly regulate transcriptional activation of *Dll4* in endothelial cells[48]. Our RNA-seq analysis identified *sox17* and the related *sox32* as HE-specific but Runx1-repressed genes. This was further validated in

*runx1−/−* mutants (Fig. 5e, f; Supplementary Fig. 8c), thus providing the first functional in vivo data on *sox17* repression in the HE. Direct RUNX1 binding to *Sox17* has been reported previously during haematopoietic differentiation in culture[49]. Together, our findings suggest a model in which different Jag1 to Dll4 ratios are established by a Runx1-dependent downregulation of *dll4* expression in the HE, either directly or indirectly via repression of *sox17* (Supplementary Fig. 8d).

**HE of *runx1−/−* mutants maintains an arterial identity.** The arterial marker *EfnB2* distinguishes arterial from venous endo-thelium before functional distinctions are recognisable[50].

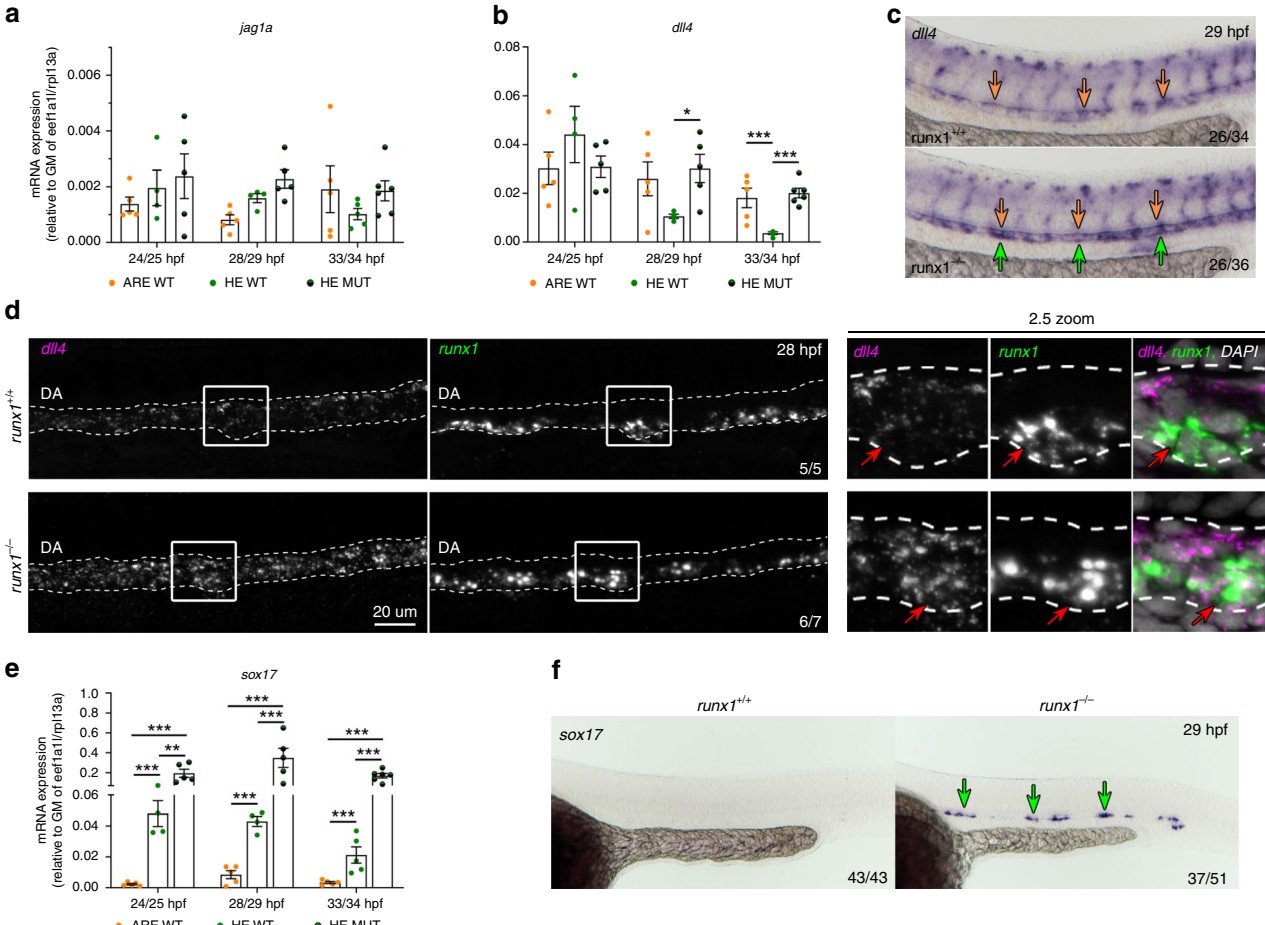

**Fig. 5** Expression of *dll4* in aortic HE is repressed by Runx1. **a, b** qRT-PCR gene expression analysis of NOTCH ligands *jag1a* and *dll4* in the HE and ARE of *runx1+/+* (WT) and the HE of *runx1−/−* mutant (MUT) embryos. Graphs in (**a**), (**b**) and (**e**) show the mean of detected expression levels relative to the geometric mean (GM) of the two housekeeping genes *eef1a1ll* and *rpl13a*. *n* = 5 independent biological experiments for WT embryos and *n* = 6 independent biological experiments for MUT embryos. Error bars represent the SEM. two-way ANOVA; *$p < 0.05$; **$p < 0.01$; ***$p < 0.001$. **c** Spatial analysis of gene expression of *dll4* in *runx1+/+* and *runx1−/−* embryos by ISH. Green arrows point to the HE. Orange arrows point to the DA roof. **d** Maximum intensity projection of representative confocal images of *runx1+/+* and *runx1−/−* embryos with double FISH for *dll4* and *runx1*. Red arrows in the close-up highlight the region of a *runx1+* cell. **e** qRT-PCR gene expression analysis of *sox17* in the HE and ARE of *runx1+/+* (WT) and the HE of *runx1−/−* mutant (MUT) embryos. **f** ISH analysis of *sox17* in *runx1+/+* and *runx1−/−* embryos. Green arrows point to the HE

Similarly to *dll4* and *sox17*, expression of *efnb2a* was increased within the HE of *runx1−/−* mutants, whereas expression of the venous marker *ephb4b* was unchanged and generally low (Fig. 6a). This prompted us to analyse expression of further arterial and venous markers in our RNA-seq data. Strikingly, arterial genes were generally upregulated in the HE of *runx1* morphants, whereas venous genes were unaffected (Fig. 6b). HE not only retained an arterial programme in the absence of Runx1 but also remained integrated into the DA lining, as detected by *runx1* expression or Citrine fluorescence (Supplementary Fig. 9a), even at 6 dpf (Fig. 6c, d).

To determine whether arterial identity is a prerequisite for aortic HE, we analysed MO-mediated repression of *dll4* and indeed found a loss of *runx1* expression (Fig. 6e), as well as of *cmyb* that could be rescued by overexpressing *runx1* in *dll4* morphants (Supplementary Fig. 9b), while endothelial integrity was unaffected (Supplementary Fig. 9c). Consistent with these data, *dll4−/−* mutants also showed loss of *runx1* expression in HE (Fig. 6f). Similarly, *Dll4−/−* mice were reported to lack Runx1 expression in the HE at E9.5 but could not be investigated at later stages due to early lethality[45]. Of note, knock-down of the arterial gene *cldn5b* also resulted in a loss of *runx1* (Supplementary Fig. 9d), further suggesting that the arterial identity is required beyond simply the Notch input. In agreement with our observations in *runx1−/−* mutants, *dll4* expression was increased in those *cldn5b*-deficient embryos. Together, these data suggest a model in which aortic HE is derived from a Dll4+ Cldn5b+ arterial progenitor population (Fig. 6g), consistent with our observation that initially most aortic cells hold the potential to upregulate *runx1P2:Citrine* expression (Supplementary Fig. 2g,h). Subsequent lumenisation creates the geometry for a signalling gradient, whereby ARE stays arterial and the DA floor commits to the haemogenic lineage[25]. The upregulation of *Runx1* then leads to the repression of arterial genes, allowing further differentiation towards the haematopoietic fate.

## Discussion

Two recent publications have concluded that HE is a distinct lineage from aortic endothelium[12,13]. However, neither paper was looking at the HE of the DA: one was studying yolk sac HE and the other HE differentiated from PSCs. Here, making use of a newly generated transgenic zebrafish line, we have profiled HE from the DA and compared it to the endothelium of the aortic roof. We find that while at earlier stages of development the gene

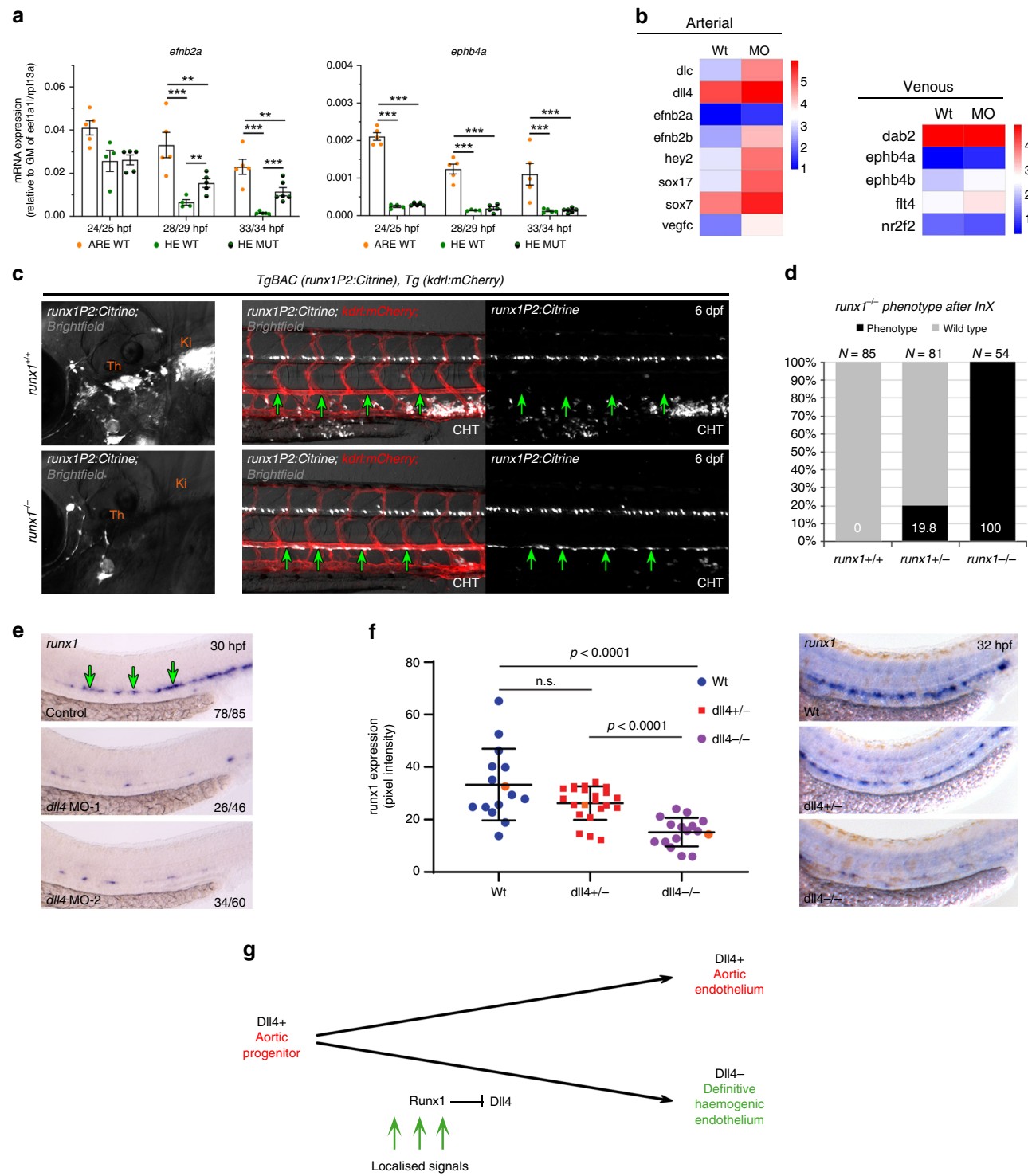

expression profiles are very similar, HE and ARE eventually become distinct. We further find that the transition is mediated by Runx1 and that, if Runx1 is inactivated, the cells that would have formed HE retain the profile of ARE, remaining in the aortic endothelium and failing to undergo EHT. We show that Runx1 is indeed acting on $dll4^+$ cells, the main marker for the non-haemogenic endothelium in the PSC study[13], by observing co-localisation of transcripts to individual DA floor cells before $dll4$ is silenced.

The HE-specific repression of $dll4$ by Runx1 additionally suggests how high Jag1/Dll4 ratios are established in the zebrafish

DA mediating low Notch1 signalling to the HE (Supplementary Fig. 5d) as detected by Gama-Norton et al.[44] in mouse. While in the Gama-Norton et al. study the sensor for strong Notch1 signalling depicted activity only in the endothelium that is distinct from HSC precursors[44], we speculate that, as in zebrafish, murine HE and aortic endothelium share a common precursor initially experiencing high Notch1 signalling through Dll4. Since the respective sensor in the Gama-Norton et al. study was induced at E10.5 and thus ~24 h after $Runx1$ expression is detectable[44], together with the repression of $Dll4$ by Runx1 that we have identified, our proposed model of a common lineage is in

**Fig. 6** Aortic HE maintains arterial features in the absence of runx1. **a** qRT-PCR gene expression analysis of the arterial marker *efnb2a* and the venous marker *ephb4b* in the HE and ARE of *runx1*[+/+] (WT) and the HE of *runx1*[−/−] mutant (MUT) embryos. Graphs show the mean of detected expression levels relative to the geometric mean (GM) of the two housekeeping genes *eef1a1l* and *rpl13a*. $n = 5$ independent biological experiments for WT embryos and $n = 6$ independent biological experiments for MUT embryos. Error bars represent the SEM. Two-way ANOVA; *$p < 0.05$; **$p < 0.01$; ***$p < 0.001$. **b** Heatmaps of whole-genome gene expression data from RNA-seq for the HE of control and *runx1* MO embryos showing arterial and venous genes. **c** Representative fluorescent microscopy image of 6 dpf *TgBAC(runx1P2:Citrine), Tg(kdrl:mCherry)* double transgenic embryos on a *runx1*[+/+] or *runx1*[−/−] genetic background. Left: Region depicting the region of definitive haematopoietic niches including the thymus (Th) and the kidney (Ki). Right: region depicting the DA and the beginning of the caudal haematopoietic tissue (CHT). Green arrows point to the ventral wall of the DA. **d** Quantification of embryos derived from the indicated in-crosses (InX) depicting the *runx1*[−/−] mutant phenotype as depicted in (**c**). **e** ISH analysis of *runx1* in control and *dll4* MO embryos. Experiment was performed with two different MOs. Green arrows point to the HE. **f** Offspring of *dll4*[+/−] heterozygous fish were analysed by ISH for *runx1* at 32 hpf. Runx1 expression was quantified by image analysis and subsequent genotyping[57]. Representative examples of runx1 expression in each of the WT, *dll4*[+/−] and *dll4*[−/−] genotypes are shown in the right-hand panels and the corresponding expression values are shown as orange points in the graph. ($\mu_{wt} = 33.3$, $\mu_{mut} = 15.19$, $p < 0.0001$, $F_{(2,50)} = 16,79$, one way ANOVA). **g** Model depicting the lineage relationship between the definitive HE giving rise to HSCs and arterial endothelium in the DA during embryogenesis. Dll4[+] aortic progenitors get exposed to localised signals patterning the DA and inducing haemogenic gene expression. Haemogenic specification includes upregulation of *runx1* expression and a subsequent Runx1 induced repression of *dll4*, thus giving rise to Dll4[−] mature HE

agreement with those observations. Furthermore, by inactivating a representative arterial gene, *dll4*, we show that the arterial state is a necessary prerequisite for HE formation in vivo. Indications of a *Runx1* dependence on Dll4 have also been reported in mice, along with *Gata2*[45], and HE dependence on the arterial programme has also been suggested using *EphrinB2*[−/−] mutants[51]. Together with the here presented imaging and gene expression data, this provides strong support for intra-embryonic HE being derived from arterial angioblasts. Lastly, a necessity for the repression of arterial identity genes, e.g. *Sox17* and *Notch1*, was shown for the endothelial to hematopoietic fate switch in the mouse DA[14]. Here, we show that the repression of the arterial programme is Runx1 driven, even though it remains to be clarified whether this repression is direct or indirectly regulated.

We propose that vascular endothelial progenitors, that separated from the primitive blood precursors in the posterior lateral plate mesoderm[52,53], become committed to a DA fate at around 13–15 hpf in zebrafish[54,55] in the absence of a distinction between future roof and floor cell fates. These precursors migrate to the midline and form a cord which starts to lumenise from 18 hpf[56]. It remains to be determined at which time point HE diverges from the arterial lineage. Runx1 is initially not required for HE differentiation as expression of a non-functional *Runx1-LacZ* reporter gene is initiated normally in the floor of the mouse DA[57], and haemogenic competence can already be detected in murine 23GFP[+] cells before *Runx1* is expressed[58]. Similarly, expression of haemogenic *gata2b* in the zebrafish DA can be detected already around 18–20 hpf[23], and it is suggested that HSC fate is partly established around 15–16 hpf during axial migration of DA precursors[59], after the arterial and venous fate decision[54]. Subsequent DA cord lumenisation results in the floor of the DA remaining proximal to inductive BMP signals and distal to Hedgehog signalling, which leads to haematopoietic gene expression, starting with *gata2b* and *runx1*[23,25], while the roof cells move away from the BMP source (Supplementary Fig 2h). That the roof cells are equivalent to floor cells in their ability to respond to BMP signalling was demonstrated by manipulating the BMP pathway using FGF inhibition, which resulted in dorsal extension of the BMP signalling domain accompanied by dorsal expression of Runx1 in the DA[60]. A previous study described a sub-population of venous-fated angioblasts in the DA[61], thus increasing the cellular complexity[61]. However, since we never observed Citrine[+] cells in the vein we suggest that these are different from HE-forming angioblasts.

Together with manipulation, both up and down, of the Notch pathway[30,31,44], these data point the way towards the successful differentiation of PSCs into HSCs. We propose that it will be crucial to first identify conditions that specifically generate endothelial cells of the aortic lineage of the trunk region[52] before inducing the differentiation into HE-like cells in vitro. Importantly, similar observations have been made in a human PSC differentiation culture, strongly supporting our in vivo data[62].

Despite its crucial role in HSC formation, how RUNX1 promotes haematopoietic cell fate is still unclear[18,20]. Here, we have identified a diverse set of potential target genes. Their respective roles in DNA methylation, signalling and niche interaction begin to reveal the mechanisms by which Runx1 transforms arterial endothelial cells into fully functional HSCs. The finding that expression of five zebrafish *Dnmt3* isoforms was substantially reduced in *runx1*[−/−] mutants highlights the importance of DNA de novo methylation for haematopoiesis[63] and expands on indications that RUNX1 partly acts through global reorganization of the epigenetic state[20,64]. DNA methylation could be stabilizing the haematopoietic programme as shown for the Dnmt3bb.1-dependent maintenance of *cmyb* expression[29]. Conversely, DNA methylation may permanently silence the endothelial programme that is initially repressed by GFI1/GFI1b and LSD1 function[65]. Recruitment of DNMT3a by LSD1 has been described[66].

*Irf1b* is a member of the inflammatory signalling pathway, recently discovered to be critical for definitive haematopoiesis[34,47]. In *irf1b* we have identified the first inflammatory gene to be downstream of Runx1 activity, the marker used for monitoring inflammatory input into HSC development. The interaction however might be indirect since *runx1*[−/−] mutants have fewer primitive neutrophils[39], which are the source of pro-inflammatory signals[47].

The Runx1 target, Pik3cd, is a member of the mTOR pathway and directly regulated by Runx1 in acute megakaryocytic leukaemia cells[67]. PI3K activity is antagonized by PTEN, which in zebrafish was shown to prevent hyperproliferation of HSPCs in the CHT[68]. Regulation of *Pik3cd* by RUNX1 might therefore be involved in fine-tuning the cell cycle of future HSCs.

Additional Runx1 targets identified here, such as *angpt1* and *mpl*, are haematopoietic niche interaction factors[41–43,69]. Of note, *Mpl*[−/−] mutant mouse embryos show an impaired survival/self-renewal of HSCs during their passage to the foetal liver[70]. It appears that RUNX1 equips the HSC with the required hardware for proper niche interaction.

Overall, we have shown that HSC-generating HE derives from aortic endothelium whose subsequent differentiation into HSPCs is dependent on the action of Runx1. We have further shown that Runx1 drives repression of the arterial endothelial programme, regulates transcription of epigenetic regulators that might be

involved in stabilising this fate switch and sets up the HSC programme including its niche-interaction capacity.

## Methods

**Zebrafish lines and maintenance**. Wild-type, $Tg(kdrl:Hsa.HRAS-mCherry)$[71] (here called $Tg(kdrl:mCherry)$), $dll4^{sa9436}$ mutants[72] (here called $dll4^{+/-}$ and $dll4^{-/-}$) and $runx1^{W84X}$ mutants[39] (here called runx1$^{+/-}$ and runx1$^{-/-}$) were bred, maintained and staged as described[22]. All zebrafish work was approved by the Research Ethics Committee of the University of Oxford.

$dll4^{sa9436}$ mutants were genotyped using a custom KASP Assay (LGC Genomics) targeting the sequence CGGCCACTACACCTGCAACCCAGATGGCC RGTTATCCTGTCTCCCTGGCT[G/A]GAAGGGGGAATACTGCGAAGAACGT AAGCAATCAGAGWTCACAATTTATT, following the manufacturer's instructions. The $TgBAC(runx1P2:Citrine)$ transgenic reporter was generated in the KCL background. $TgBAC(runx1P2:Citrine);Tg(kdrl:mCherry)$ double transgenic, $TgBAC(runx1P2:Citrine);runx1^{-/-}$ and $Tg(kdrl:mCherry);runx1^{-/-}$ animals were generated by natural mating. Mutant embryos were identified by genotyping fin-clipped 3.5 dpf embryos. DNA was extracted using the HotSHOT method[73], PCR performed using W84X genotyping primers (Supplementary Data 5) and diagnostic gels were run after *HaeII* digest.

**Morpholino and DNA injections**. Antisense MOs (GeneTools) were used to target $runx1$[22], $gata2b$[23], $tal1b$[74] $cldn5b$ and $dll4$ (MO-1[75] and MO-2[76]) at the amounts indicated (Supplementary Data 5). Typically, 0.5–1 nl total volume of MO was injected in 1–2 cell stage embryos. For rescue experiments, full length Runx1 was amplified from cDNA of 28 hpf embryos and a flag-tag added to the 5′ end. This fragment was cloned downstream of the Ef1alpha promoter and a Rabbit β1-globin intron B type 2 allele (accession number J00660 (552–1196)) and upstream of a SV40 polyA sequence, flanked by miniTOL2 sites. Fifty picograms of plasmid DNA was injected with 25 pg transposase mRNA for the runx1 overexpression experiment.

**BAC cloning and transgenesis**. BAC recombineering was based on the Suster et al. protocol[77] with the following alterations. The $Citrine-SV40pAFKF$ (containing *kanamycin* flanked by two FRT sites, a kind gift from Prof. Tatjana Sauka-Spengler) transgene cassette was introduced before the iTol2-amp cassette. BAC recombineering was performed in SW105 cells. Amplification of the homology-constructs with 50 bp overhangs was performed with specific primers (Supplementary Data 5). Recombineered BACs were purified using the Invitrogen Pure-Link HiPure Plasmid Midiprep Kit (cat no K2100–04). Embryos were injected with 1 nl of DNA (180 ng/µl) plus *tol2* mRNA (120 ng/µl) into the cell of early 1-cell stage embryos.

**Whole mount ISH**. Whole-mount in situ hybridization was carried out as described[22]. cDNA templates for in situ hybridization probes were generated by PCR amplification from 20–28 hpf embryo cDNA (see Supplementary Data 5 for primers) and cloned into pGEMT-Easy. Probes for *kdrl*, *runx1*, *cmyb* and *dll4* are described elsewhere[16,28,46,78]. Analysis and quantification of *runx1* gene expression after in situ hybridization in *dll4* mutants was performed blinded. Briefly, we imaged all embryos from a $dll4^{+/-}$ incross, inverted and converted the images to 8-bit grayscale. Next, we measured pixel intensity in an ROI containing the *runx1* staining against an equivalent unstained background ROI to obtain a pixel intensity value for each embryo[79]. After genotyping using the recommended KASP assay (LGC Genomics) primers for the *dll4* mutation[72] following the manufacturer's instructions, pixel intensity values were assigned to each image and grouped according to genotype. The mean intensity values were plotted in a dot plot and analysis of the differences between genotypes was performed (Graphpad 8.0 software, one way ANOVA).

**FISH and immunohistochemistry**. Double fluorescent in situ hybridization was performed as previously described[80]. Briefly, embryos were incubated at 65 °C for 18 h in 2 ng/µl DIG- and fluorescein-labelled mRNA probes *runx1*, *Citrine* (*GFP*) and *dll4*. After washes, the embryos were sequentially incubated with horseradish peroxidase-conjugated anti-FLUO (1:500; Roche) and anti-DIG (1:500; Roche) antibodies. The RNA probes were sequentially developed with custom-made FLUO (FITC; 1:200) and tetramethylrhodamine (TAMRA; 1:100) conjugated tyramide in PBST containing 0.003% (v/v) $H_2O_2$ for 35 min at RT.

For FISH combined with immunohistochemistry first FISH was performed according to the standard in situ hybridisation protocol. The signal was developed with SIGMAFAST Fast Red TR/Naphtol. Embryos were rinsed in phosphate-buffered saline with Tween20 (PBT) and directly processed for immunohistochemistry. Embryos were blocked in blocking buffer (5% goat serum/ 0.3% Triton X-100 in PBT) for 1 h at RT before incubated with primary antibody against GFP (rabbit, 1:500, Molecular Probes), diluted in blocking buffer overnight at 4 °C. Secondary antibody raised in goat coupled to AlexaFluor488 (Invitrogen) was used in 1:500 dilutions for 3 h at RT. Hoechst 33342 was used as a nuclear counterstain.

Fluorescent images were taken on a Zeiss LSM880 confocal microscope using 10× air, 40× oil or 63× oil immersion objectives. Images were processed using the ZEN software (Zeiss) to generate maximum intensity projections.

**Wide-field fluorescence and live confocal microscopy**. For wide-field fluorescence microscopy live embryos were anaesthetised with 160 µg/ml MS222 and mounted in 3% methylcellulose. Embryos were assessed and imaged on a SteREO Lumar V12 microscope (Zeiss) with an AxioCam MRm (Zeiss) or on a MVX10 stereo microscope (Olympus) with an ORCA-Flash4.0 LT Digital CMOS camera. Image acquisition and analysis was conducted with the AxioVision software (Zeiss).

For live confocal microscopy embryos were grown in E3 medium supplemented with PTU to prevent pigment cell formation. For imaging, embryos were anaesthetised with 160 µg/ml MS222 and orientated at the bottom of 35 mm glass bottomed dishes in 1% low melt agarose. Embryos were covered with E3 media containing MS222 and imaged on an LSM780 or 880 confocal (Zeiss) at 28.5 °C. Images were processed using the ZEN software (Zeiss) to generate maximum intensity projections.

**Cell dissociation and fluorescent activated cell sorting**. Embryos were staged accordingly, pre-sorted for the presence of fluorescent reporters using the Olympus stereo microscope MVX10 and collected in low binding microcentrifuge tubes (SafeSeal Microcentriguge Tubes, Sorenson; Cat#39640T). Yolk was removed using 116 mM NaCl/2.9 mM KCl/5 mM HEPES (with freshly added 1 mM EDTA) deyolking buffer. Cells were dissociated using collagenase/trypsin buffer (20 mg collagenase in 0.05% Trypsin with EDTA in 1× HBSS solution). Reaction was stopped in 1× HBSS/10 mM HEPES/0.25% BSA. Dissociated cells were passed through a 40 µm cell strainer and re-suspended in appropriated volume (~10 µl per embryo; ~3–7 × $10^6$ cells per ml) of 1× HBSS/10 mM HEPES/0.25% BSA with Hoechst$^{DEAD}$ (33258; Invitrogen) (1:4000 dilution). FAC-sorting was carried out by the WIMM Flow Cytometry Facility using the *BD FACS Aria Fusion* system. To identify the gating strategy for the DP-cells, we split the DP-gate into three sub-gates of similar height in order to select for a DP-R1$^{hi}$ population with best enrichment of haemogenic marker genes. Cells were either directly sorted into RLT buffer (RNA isolation) or into 1× HBSS/10 mM HEPES/0.25% BSA buffer for subsequent preparation of ATAC-seq libraries.

**RNA extraction from FAC-sorted cells**. To isolate RNA for qRT-PCR or RNA-sequencing, cells were sorted into 350 µl RLT buffer (QIAGEN) containing 20 µl β-mercaptoethanol per 1 ml RLT buffer and adjusted to a volume of 450 µl with RNA free $H_2O$ (each sorted cell equals a volume of ~0.00325 µl). The cell lysate was thoroughly vortexed and run through a QIAshredder (QIAGEN) column. RNA was cleaned up using the RNeasy Plus Micro Kit (QIAGEN) according to manufacturer's protocol. RNA was eluted twice with 10 µl of RNA free $H_2O$. RNA quality and quantity was determined using the 2100 Bioanalyzer Pico kit (Agilent technologies).

**qRT-PCR**. cDNA was synthesized from total RNA (2–5 ng) using a SuperScript IV (ThermoFisher) enzyme according to manufacturer's instructions in a total volume of 20 µl using random hexamer primers (250 ng) and 10 mM dNTPs. qRT-PCR was performed on a 7500 real-time cycler (Applied Biosystems) using SYBR Green (Fast SYBR Green; Applied Biosystems). Experiments were carried out in technical duplicates with four individual biological replicates. qRT-PCR primers are listed in Supplementary Data 5. Fold changes in gene expression were calculated using the $2^{-\Delta\Delta C\tau}$ method and normalized to a geometric mean of two housekeeping genes, *ubc* and *eef1a1l1*.

**Multiplexed gene expression analysis**. Multiplex qRT–PCR was performed using the Fluidigm (BioMark) platform. TaqMan probes (ThermoFischer) (see Supplementary Data 5) were pooled to a concentration of 0.2×. Before cell sorting, an RT-PreAmp master mix (consisting of 5 µl 2× Reaction Buffer and 1.2 µl RT/Taq enzyme (ThermoFischer SuperScript III One-Step RT-PCR System with Platinum Taq kit), 0.1 µl SUPERase-In RNAse Inhibitor (Ambion), 1.2 µl TE buffer (Invitrogen), and 2.5 µl of 0.2× TaqMan assay mix) was freshly prepared on the day. All pipetting steps were performed in a clean-room.

Hundred cells were directly sorted by FACS into 10 µl of RT-PreAmp master mix. Collected samples were immediately vortexed and spun to aid cell lysis. RT-PCR (15 min at 50 °C; 2 min at 95 °C) and pre-amplification (15 s at 95 °C and 4 min at 60 °C; 20 cycles) was performed in a PCR machine and diluted with 40 µl TE buffer. The pre-amplified mix was immediately frozen at −20 °C until further analysis. Gene expression analysis was performed using a 48.48 IFC chip (Fluidigm) according to the manufacturer's instructions. Ct thresholds were set for each assay with the same thresholds used across all experiments. Ct values were calculated by BioMark Real-time PCR Analysis software (Fluidigm). Data were exported to Excel as.csv files for subsequent analysis.

**RNA sequencing analysis**. At least 300 $TgBAC(runx1P2:Citrine);Tg(kdrl: mCherry)$ double transgenic embryos for both *runx1* MO and wild-type conditions

were processed for RNA isolation from specific cell populations after FAC-sorting. To control for batch effects potentially derived from different clutches, every clutch of embryos was split in half, whereby one half remained un-manipulated and the other was injected with *runx1* MO. Injected and non-injected embryos were pooled accordingly and staged to 28–30 hpf for subsequent FAC-sorting. RNA-seq library generation (SMARTer libraries for low-input RNA) and sequencing (Illumina HiSeq4000 with 75 bp paired end) was performed by the Welcome Trust Centre for Human Genetics (WTCHG) using 2.2 ng of total RNA. ~24–30 million reads per sample were generated.

Sequenced reads were checked for base quality, trimmed where 20% of the bases were below quality score 20, and filtered to exclude adapters using Trimmomatic (Version 0.32). Sequences were aligned to the Zebrafish Genome Zv10 with STAR with default parameters. Aligned read features were counted using Subread tool: featureCounts method (version 1.4.5-p1). Differential gene expression (DEG) analysis was carried out using EdgR (Bioconductor Package).

To determine number of mapped reads we used the trimmed data. The alignment has been performed using STAR with default parameters. The number of mapped reads (QC-passed reads count) has been obtain using Samtools mapping statistics (flagstat tool).

For ANOVA-like test analysis, the generalized linear model (GLM) analysis was used as followed: Estimates "Dispersion", fitted with "negative binomial model" and estimates "Generalized linear model likelihood ratio". Genes with a False Discovery Rate (FDR) less than 0.05 were filtered out. Further, genes were filtered for log2 Fold Change < −0.7 and >0.7. Reads per kilobase per million (RPKM) values were used as input for Clustering analysis. A z-score transformation of the log2 replicates mean for each sample was performed. Clustering analysis was performed using the Consensus Clustering Plus algorithm[81] with maximum cluster number to evaluate equal to 15. The algorithm generated a CDF plot assessing number of eight clusters as best stability. Using the genes of each cluster, heat-maps were generated using the z-score values.

To identify genes that were differentially expressed between two populations, we used GLM analysis. For the DP-R1lo vs. DP-R1hi DEG analysis all endothelial populations (SP-kdrl, DP-R1lo, DP-R1med and DP-R1hi) were included into ANOVA-like test analysis. For the control embryo vs. MO embryo DEG analysis (for both DN and DP-R1hi) all four populations (DN-Wt, DN-MO, DP-R1hi-Wt and DP-R1hi-MO) were included into ANOVA-like test analysis. After normalization all genes with less than three samples with read counts >0 were removed prior to DEG analysis. All genes with a FDR value >0.05 or with at least three samples equal to 0 read counts were removed from further analysis. Enrichment of GO terms and KEGG pathways were calculated with DAVID6.8. Representative GO terms were taken from the BP_DIRECT class.

**ATAC-sequencing analysis**. For genome wide open chromatin analysis by ATAC-seq, 2000–3000 cells were FAC-sorted directly into 100 µl of 1× HBSS/10 mM HEPES/0.25% BSA buffer and spun down at 500 g for 5 min at 4 °C. Tagmentation was performed as described[82], but with 1.5 µl Tn5 transposase (Illumina) in a 50 µl reaction volume. Immediately after transposition, DNA was purified using the QIAquick PCR purification kit (Qiagen), eluted 2× with 10 µl EB buffer and stored at −20 °C. Eighteen microlitres of transposed DNA was taken forward into a 50 µl PCR reaction mix with barcoded primers and 25 µl of 2× NEBNext High-Fidelity 2× PCR Master Mix (NEB). Fragments were amplified for 16 cycles.

As a genomic DNA control, DNA from 30,000 cells was isolated using the DNeasy Blood & Tissue kit (Qiagen) and eluted with 25 µl EB buffer. Tagmentation was performed as described. Fragments were amplified for nine cycles. Fragment sizes were verified using Tapestation (Agilent) and libraries were quantified using KAPA Library Quant Kit for Illumina Sequencing Platforms (KAPABiosystems). Sequencing was performed on a NextSeq machine.

Sequenced reads were checked for base qualities, trimmed where 20% of the bases were below quality score 20, and filtered to exclude adapters using Trimmomatic (Version 0.32). Sequences were aligned to the Zebrafish Genome Zv10 with BWA (Version 0.7.12) with default parameters. Aligned read features were counted using Subread tool: featureCounts method (version 1.4.5-p1). Peaks were identified using MAC2 software (Version 1.4.2), using the genomic DNA as input to define background. Differential peaks analysis was carried out using DiffBind (Bioconductor Package). Triplicates were analysed for consistency by PCA and correlation analysis and one outlying replicate has been removed for each, DP-R1$^{hi}$ and DP-R1$^{lo}$. Using a GLM analysis with EdgeR methods a list of DE packs between DP-R1$^{hi}$ and DP-R1$^{lo}$ population has been identified, then annotated with Homer software (Version 3.0). Further, peaks were filtered for log2 Fold Change < −0.7 and >0.7. Selecting the specific peaks for DP-R1$^{hi}$ and DP-R1$^{lo}$. "De novo analysis" for transcription factor binding site has been carried out using the TSS, intergenic, and intron peaks list with "findMotifsGenome.pl" in Homer software package.

**Reporting summary**. Further information on research design is available in the Nature Research Reporting Summary linked to this article.

## Data availability

All data generated or analysed during this study are included in this published article (and its supplementary information files). Accession codes for RNAseq and ATACseq have been deposited in the GEO database under accession codes: GSE132259 and GSE132258, respectively.

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

## Acknowledgements

We thank Paul Liu for providing the W84X line. We thank the Crosier lab for providing the 14i20, 74j23, 97a02, and 135g16 zebrafish BACs containing the *runx1* locus. We thank Tatjana Sauka-Spengler for the the BAC-recombineering protocol and the provision of the *Citrine-SV40pAFKF* cassette. We are very grateful to the staff of the Biomedical Services Unit for excellent fish husbandry. We thank Kevin Clark from the WIMM flow cytometry facility for cell sorting. The flow cytometry facility is supported by the MRC HIU, MRC MHU (MC_UU_12009), NIHR Oxford BRC and John Fell Fund (131/030 and 101/517), the EPA fund (CF182 and CF170), and WIMM Strategic Alliance awards G0902418 and MC_UU_12025. We thank Christoffer Lagerholm from the Wolfson Imaging Centre Oxford for help with imaging. The Wolfson Imaging Centre Oxford is supported by the MRC via the WIMM Strategic Alliance (G0902418), the Molecular Haematology Unit (MC_UU_12009), the Human Immunology Unit (MC_UU_12010), the Wolfson Foundation (grant 18272), and an MRC/BBSRC/EPSRC grant (MR/K015777X/1) to MICA – Nanoscopy Oxford (NanO): Novel Super-resolution Imaging Applied to Biomedical Sciences, Micron (107457/Z/15Z). The facility was supported by WIMM Strategic Alliance awards G0902418 and MC_UU_12025. We thank Emmanouela Repapi from the CBRG for advices on the sequencing analysis. This research was supported by the MRC MHU programme number MC_UU_12009/8 and MRC MHU studentship MC_UU_12009. The

work of R.M. and M.K. is funded by the British Heart Foundation (BHF Oxford CoRE Fellowship, BHF IBSR Fellowship FS/13/50/30436).

## Author contributions

F.B., P.P., T.P., M.D.B., R.M. and R.P. all participated in the design of the study. F.B. performed most experiments and analyzed the data. R.R. performed the bioinformatics analyses. P.P., T.P., M.K., J.S., R.M. and I.H.C.T. performed experiments, F.B. wrote the paper and F.B., R.M., T.P. and R.P. edited the paper.

## Additional information

**Competing interests:** The authors declare no competing interests.

