## [Transparent peer review file · Nature Communications]

Reviewers' Comments:

Reviewer #1:

Remarks to the Author:

SUMMARY and RECOMMENDATION

Bonkhofer et al. present findings supporting the idea that hemogenic endothelium derives from an arterial endothelial precursor, and involves a binary choice between arterial endothelial fate and HSC precursor fate. In their evidence-supported model, Dll4 upregulation from a basal level in some cells of the primitive arterial endothelium leads to inhibition of runx1 and the downstream definitive hematopoietic program, or alternatively, Runx1 inhibits Notch signaling and activates the hematopoietic program. They further identify Runx1 transcriptional targets and provide evidence that arterial specification is a pre-requisite to definitive hematopoietic potential. The work is highly convincing, with clear results that are clearly presented. Figures are well laid out, with informative and helpful schematics. The writing is lucid. The questions of whether arterial and hemogenic endothelium are specified as discrete pools during embryogenesis remains controversial, and this work provides crucial insight into this question, as well as helping to inform the outstanding question of whether arterial specification is a precondition to hematopoietic potential. We believe the manuscript is of high quality, addressing important questions, with many useful findings, which will be of high interest to researchers in the field and in general. We believe it could be suitable for publication by addressing the minor issues presented below.

MINOR

- A major point of the manuscript is that hemogenic endothelium represents a differentiated state of arterial endothelium related by lineage to a shared precursor in the primitive endothelium of the lumenized dorsal aorta or primitive vascular cord. The overall model is that second wave P2 runx1 expression in the DA drives inhibition of dll4 (and other genes), which when expressed would lead to the alternative arterial endothelial fate. A key issue is when future HSCs are specified. Is it before or after formation of the primitive vascular cord? Before would be consistent with some of the published findings putatively refuted here, such as those of Frame et al. and Ditadi et al. The authors of the current manuscript suggest (and provide some evidence) that choice between hematopoietic and endothelial fate is after formation of the primitive vascular cord or arterial endothelium. Evidence supporting this idea includes, especially, the gene profiling studies and the redirections of fate potential in Dll4 knockdown studies. A weak point is the explanation of Citrine+ cells in the (runx1-transcript negative) ARE, ISVs, and primitive erythrocytes. They speculate that early expression of Citrine (at 11-12 hpf) in the LPM is followed by a new burst of definitive runx1 (and reporter) expression in (some of?) the descendant arterial endothelial cells. However, the Citrine observations are equally consistent with the possibility that HSC fate is specified much earlier, and the capability of second wave definitive runx1 (and reporter) expression in the arterial endothelium is a feature of this prior specification. We agree that the studies presented are much more consistent with authors' model, but believe that some of the caveats should be presented. Overall, it would be helpful to much more clearly and explicitly articulate the model for the exact progression of runx1 and Citrine reporter acquisition, loss, and amplification, and connect it to the cell-sort populations and their putative fate potentials at an earlier point in the text. What is the source of runx1P2:Citrine fully negative cells in the dorsal aorta at 24 hpf (i.e. Kdr1-SP cells)? What is the exact trajectory of maturation of DP-R1lo, med, and hi populations? Do the authors think cells face a binary choice between endothelial and hematopoietic fate at a particular, definable point, and if so, when is it? Do all arterial endothelial cells begin as runx1P2:Citrine-lo, and then either up- or downregulate runx1? Only some of them? How does this work?

The model is further (explicitly) predicated on the notion that arterial and venous fates are specified very early (i.e. earlier than 18-20 hpf). From the Conclusions: "We propose that the dorsal lateral plate mesoderm" [what is the meaning of 'dorsal' lateral plate mesoderm in the zebrafish—do they mean 'posterior' lateral plate mesoderm?] "becomes committed to a dorsal aorta fate at around 13-15 hpf in zebrafish in the absence of a distinction between future roof and floor cell fates." The schematic in Fig. S2h also depicts this putative early segregation. We agree

that there is substantial evidence supporting this model, for example, the referenced Zhong TP et al., *Nature* 2001, as well as the unreferenced Kohli V et al., *Dev Cell* 2013. However, there is also evidence supporting the idea that the primitive vascular cord represents an unsegregated mixed population of endothelial lineage potentials, for example, Herbert SP et al., *Science* 2009. We believe this controversy should be addressed, either by acknowledging it, or stating why apparently conflicting results are not in conflict.

- Kdr1 transgenics label non-endothelial tissues. For example, double transgenic kdr1 and fli1a reporters have kdr1 single positive cells. Kdr1-SP cells are therefore not a perfect NAE control, because this population includes undefined, non-endothelial cells. The authors RT-qPCR profiling should therefore include internal controls, including runx1, kdr1, Citrine, and mCherry, as well as specificity controls, like fli1a and cdh5 (endothelium), myoda (myotome), cdh17 (pronephros), pdgfra (sclerotome and neural crest), and dab2 (venous endothelium).
- The authors argue that high expression of dll4 in the DP-R1lo population indicates its arterial identity, but SP-kdr1 cells show the highest level efnb2a expression, which would imply these are the most "arterial." The idea that DP-R1lo cells are artery and the SP-kdr1 cells are venous endothelium (as depicted in the schematic) would explain dll4 levels being highest in the DP-R1lo population, but then why are efnb2a levels highest in this "non-aortic endothelium"?
- We agree that there is strong evidence, including presented in this manuscript, that being arterial endothelium is a pre-requisite to HSC potential; however, the statement, "we show that the arterial state is a necessary prerequisite for HE formation," based solely on the fact that dll4 KD represses arterial and HSC fate, is an overstatement. We note that it is—in highly artificial circumstances—possible to achieve expression of HE/HSC precursor markers in the absence of arterial markers, for example, in Figs. 5 and 6 of Ren, Gomez, Zhang, and Lin, *Blood*, 2010.
- There are curious and conspicuous omissions in the references. For example, the authors reference Kissa & Herbomel, 2010 and Boisset et al., 2010 in support of the statement, "During embryogenesis, HSCs are generated from an endothelial precursor termed haemogenic endothelium (HE)," but not Bertrand et al., 2010, which appeared as trio with the other two articles in the same issue of *Nature*? The authors cite Ciau-Uitz & Patient, 2000, North et al., 1999, and Gering and Patient 2005 in support of the statement, "The transcription factor Runx1 is expressed in HE and is essential for the emergence of HSCs," but not Kalev-Zylinska et al., 2002 and Chen et al., 2009, which are arguably the seminal functional studies on requirement for Runx1 in zebrafish and mouse respectively. As the authors' expertise in this field is beyond question, we would suggest re-inspecting the referencing of claims to assure that statements are referenced either to reviews, or—if to seminal works—then to the complete sets of seminal works.

Reviewer #2:

Remarks to the Author:

The manuscript entitled " Blood stem cell forming haemogenic endothelium derives from arterial endothelium" by Bonkhofer et al utilizes the zebrafish model to examine lineage relationships between the endothelial populations in the embryonic dorsal aorta. The authors develop a novel Runx1 reporter that highlights the earliest onset of Runx1 expression in the aorta in a fairly specific fashion, and go on to characterize cells distinguished in that line in combination with a vascular reporter in both an unbiased and targeted fashion. Large scale profiling analyses indicate that the Runx1+ hemogenic endothelial population is distinct from that of the roof of the aorta, as well as the non-aortic endothelium. This assay also reveals a number of previously uncharacterized factors that seem to be associated with hemogenic fate, including metabolic, inflammatory and transcriptional regulatory factors. Analysis of gradations of reporter gene expression show that the higher the level of Runx1, the lower the level of arterial associated markers. However, Runx1 reporter expression is consistently associated and indeed appears to be dependent on expression

of the arterial marker *dll4*. The authors conclude that HE cells transit through an arterial intermediate, and then down regulate this program via Runx1-mediated repression, however, they do not investigate the necessity of expression of other arterial markers nor the impact of *dll4* over-expression in a population that wasn't normally hemogenic, and thus cannot completely distinguish if there is simply a requirement for Notch signaling in HE, which is also known to be essential to arterial commitment, vs the need for an intermediate state. Despite that caveat, this is a comprehensive study of the transcriptional landscape of each of these associated populations, in vivo, including how they are being modified over time, which is of significant interest to the field.

Major issues:

1) The authors make an intriguing case for transit through a *dll4*+ arterial intermediate during the commitment to HE fate. However, both *ephrinB2* and *dll4* are Notch targets which had been previously shown to function upstream of *gata2* to control Runx1 expression. Are there any non-Notch regulated arterial genes that could be used to confirm this isn't just coincidental necessity of Notch for two separate populations rather than a transition through an arterial intermediate?

2) A related question arises regarding *dll4* itself. The authors show that *dll4* is needed to express *runx1* in the hemogenic population, however, it would be interesting to know if it is just the Notch signaling aspect that is required, or if you truly have to be an artery. One way to test this would be enforced expression of *dll4* in a population that wasn't normally hemogenic (or one that doesn't become hemogenic until a later time point).

3) The authors do a very nice characterization of expression in the low/med/hi *runx1* population at 29hpf. Since their marker comes on earlier, and since they think HE transitions through an arterial state, if they did the same expression analysis, would they find more arterial markers in the highest *runx1* population, suggesting this is a self-reinforcing continuum?

Minor issues:

1) While the transcriptional associations are interesting, it is unclear how the low/med/hi populations were segregated in the FACS plots due to the thickness of the gating boxes. Are they discrete populations? Or was this an arbitrary fractionation of a continuum.

2) Would forced *sox17* expression block HE commitment and/or arterial program down regulation even in the presence of wt Runx1?

Reviewer #3:

Remarks to the Author:

In these studies, the authors create a Runx1 zebrafish reporter line that can presumably be used to identify and isolate hemogenic and arterial endothelial cells, which they say generate hemogenic endothelial cells. They also use this line to identify Runx1 target genes that may be involved in endothelial to hematopoietic transition (EHT). There are several significant concerns with the studies, as outlined below.

1. There are sites in the developing mouse where hemogenic endothelial cells are specified from primordial endothelial cells, and this process does not involve arterial specification (i.e. yolk sac, placenta). Therefore, the suppression of arterial phenotype is likely specific to the AGM and possibly not relevant to cells in culture, which lack spatial context and microenvironment. This should be considered and discussed with regard to the generation of hemogenic endothelial cells from stem cells in culture, and HSC therefrom.

2. In Figure 1, the expression of Citrine is more intense and more broadly distributed than endogenous Runx1. This is explained as higher protein stability of the reporter, but what is the

effect of Runx1 expression in aortic endothelial cells and intersomitic vessels that do not generate HSC?

3. In a related issue, Runx1 is not a “master” inducer of hemogenic endothelial cells (nor a specific marker) in zebrafish (as also determined in mouse by several groups) or the other endothelial cells that express it would generate HSC. This important issue should be addressed, especially since hemogenic endothelial cells are defined as Runx1+, and sorted based on Runx1 expression. Clearly, other endothelial cells express Runx1, and Runx1 expression does not induce a hemogenic phenotype.

4. There is no evidence that Runx1+ endothelial cells give rise to Runx1- endothelial cells, so this suggestion should be excluded from the Results sections.

5. These studies suggest that Runx1 regulates Notch signaling, but does Notch signaling directly regulate Runx1 expression in the zebrafish aorta? In mouse yolk sac and aortic hemogenic endothelial cells, Runx1 is upregulated downstream of Notch signaling. In addition, this occurs downstream of retinoic acid signaling and cKit expression, so what is the temporal status of cKit expression and retinoic acid signaling, relative to Notch activation, during zebrafish aorta definitive hematopoiesis?

6. There is no direct evidence presented to show that Runx1 represses arterial gene expression, and this is a main conclusion.

7. The use of morpholinos to suppress gene expression, especially without MO + gene rescue controls, is no longer considered an acceptable approach due to potential off-target effects.

8. In the ATAC analysis, the “hemogenic endothelial cells” are likely enriched for open sites related to Runx1 regulation because the analyzed cells were sorted based on Runx1 expression. Also, there are no mechanistic studies to demonstrate the functionality of putative Runx1 targets that may play a role in EHT.

Reviewers' comments:

Reviewer #1 (Remarks to the Author):

SUMMARY and RECOMMENDATION

Bonkhofer et al. present findings supporting the idea that hemogenic endothelium derives from an arterial endothelial precursor, and involves a binary choice between arterial endothelial fate and HSC precursor fate. In their evidence-supported model, Dll4 upregulation from a basal level in some cells of the primitive arterial endothelium leads to inhibition of runx1 and the downstream definitive hematopoietic program, or alternatively, Runx1 inhibits Notch signaling and activates the hematopoietic program. They further identify Runx1 transcriptional targets and provide evidence that arterial specification is a pre-requisite to definitive hematopoietic potential. The work is highly convincing, with clear results that are clearly presented. Figures are well laid out, with informative and helpful schematics. The writing is lucid. The questions of whether arterial and hemogenic endothelium are specified as discrete pools during embryogenesis remains controversial, and this work provides crucial insight into this question, as well as helping to inform the outstanding question of whether arterial specification is a precondition to hematopoietic potential. We believe the manuscript is of high quality, addressing important questions, with many useful findings, which will be of high interest to researchers in the field and in general. We believe it could be suitable for publication by addressing the minor issues presented below.

We are pleased that the Reviewer thinks that the work is highly convincing and provides crucial insight into important questions. It is also gratifying that he or she finds the manuscript of high quality and only has minor issues.

MINOR

1) A major point of the manuscript is that hemogenic endothelium represents a differentiated state of arterial endothelium related by lineage to a shared precursor in the primitive endothelium of the lumenized dorsal aorta or primitive vascular cord. The overall model is that second wave P2 runx1 expression in the DA drives inhibition of dll4 (and other genes), which when expressed would lead to the alternative arterial endothelial fate. A key issue is when future HSCs are specified. Is it before or after formation of the primitive vascular cord? Before would be consistent with some of the published findings putatively refuted here, such as those of Frame et al. and Ditadi et al. The authors of the current manuscript suggest (and provide some evidence) that choice between hematopoietic and endothelial fate is after formation of the primitive vascular cord or arterial endothelium. Evidence supporting this idea includes, especially, the gene profiling studies and the redirections of fate potential in Dll4 knockdown studies. A weak point is the explanation of Citrine+ cells in the (runx1-transcript negative) ARE, ISVs, and primitive erythrocytes. They speculate that early expression of Citrine (at 11-12 hpf) in the LPM is followed by a new burst of definitive runx1 (and reporter) expression in (some of?) the descendant arterial endothelial cells. However, the Citrine observations are equally consistent with the possibility that HSC fate is specified much earlier, and the capability of second wave definitive runx1 (and reporter) expression in the arterial endothelium is a feature of this prior specification.

We agree that the studies presented are much more consistent with authors' model, but believe that some of the caveats should be presented. Overall, it would be helpful to much more clearly and explicitly articulate the model for the exact progression of runx1 and Citrine reporter acquisition, loss,

and amplification, and connect it to the cell-sort populations and their putative fate potentials at an earlier point in the text.

What is the source of *runx1P2:Citrine* fully negative cells in the dorsal aorta at 24 hpf (i.e. *Kdr1*-SP cells)? What is the exact trajectory of maturation of DP-R1_{lo}, med, and hi populations? Do the authors think cells face a binary choice between endothelial and hematopoietic fate at a particular, definable point, and if so, when is it?

Do all arterial endothelial cells begin as *runx1P2:Citrine*-_{lo}, and then either up- or downregulate *runx1*? Only some of them? How does this work?

We thank the reviewer for pointing out the need for clarification. We have changed the wording to clearly indicate that the initial *runx1* expression around 11-12 hpf is in precursors to both endothelium and primitive blood (potentially the haemangioblast). This expression is maintained in primitive blood derivatives. However, the later *runx1* expression in the dorsal aorta HE is *de novo* expression within that specific cell lineage. To clarify this, we would like to point the Reviewer's attention to Supplementary Figure 2e, in which endogenous *runx1*, identified by ISH, can be detected in the primitive blood (eryP) at 18 hpf but not in the region of the DA until 24 hpf.

We have further added a model of *runx1* expression in the forming DA describing the trajectory of the different DA cells. We think that the aortic precursors harbour a haematopoietic potential that will be activated by exposure to inductive extrinsic signals like BMP next to the floor of the DA. Initial expression of *runx1P2:Citrine* appears to be activated initially in most DA cells before DA lumenization (see SupFig. 2h), but the subsequent positioning of the aortic cells after DA lumenization will determine if *runx1P2:Citrine* expression will be maintained and endogenous *runx1* will be up-regulated (DP-R1_{high} in the DA floor), or if it will be switched off to maintain an arterial state (DP-R1_{low} in the DA-roof).

We would like to point out that while we indeed partly refute the model by Ditadi et al., we understand that the data presented by Frame et al. as well as by Ditadi et al. are in line with our *in vivo* model, which puts the emphasis on discriminating between 2nd wave HE in the YS (with the potential to generate a lymphoid outcome) and the intra-embryonic 3rd wave HE (with the potential to generate HSCs). Our work, and the work of Frame et al., indicates that the arterial lineage might be a key difference.

2) The model is further (explicitly) predicated on the notion that arterial and venous fates are specified very early (i.e. earlier than 18-20 hpf). From the Conclusions: "We propose that the dorsal lateral plate mesoderm" [what is the meaning of 'dorsal' lateral plate mesoderm in the zebrafish—do they mean 'posterior' lateral plate mesoderm?] "becomes committed to a dorsal aorta fate at around 13-15 hpf in zebrafish in the absence of a distinction between future roof and floor cell fates." The schematic in Fig. S2h also depicts this putative early segregation. We agree that there is substantial evidence supporting this model, for example, the referenced Zhong TP et al., Nature 2001, as well as the unreferenced Kohli V et al., Dev Cell 2013.

However, there is also evidence supporting the idea that the primitive vascular cord represents an unsegregated mixed population of endothelial lineage potentials, for example, Herbert SP et al., Science 2009. We believe this controversy should be addressed, either by acknowledging it, or stating why apparently conflicting results are not in conflict.

We thank the Reviewer for pointing out the misleading wording of "dorsal" lateral plate mesoderm. We have now corrected it to "posterior".

We also thank the Reviewer for mentioning further studies that need to be cited in this context and now acknowledge the Herbert SP et al., Science 2009 study in our discussion.

3) *Kdr1* transgenics label non-endothelial tissues. For example, double transgenic *kdrl* and *fli1a* reporters have *kdrl* single positive cells. *Kdr1*-SP cells are therefore not a perfect NAE control, because this population includes undefined, non-endothelial cells.

The authors RT-qPCR profiling should therefore include internal controls, including *runx1*, *kdrl*, Citrine, and mCherry, as well as specificity controls, like *fli1a* and *cdh5* (endothelium), *myoda* (myotome), *cdh17* (pronephros), *pdgfra* (sclerotome and neural crest), and *dab2* (venous endothelium).

We thank the Reviewer for these comments. Whilst the *Kdr1*-SP is not a perfect NAE control, we feel it is appropriate to include comparison with this population. In addition though, we have now included RNAseq data for the requested internal control genes, since the sample generation was identical to the qPCR protocol. The new data is presented in Supplementary Fig. 4. We would like to mention that, in zebrafish, *dab2* is not exclusively expressed in veins at around 24-30 hpf; ISH clearly shows *dab2* expression in the roof of the DA. However, we have included *nr2f2* as a vein marker.

4) We agree that there is strong evidence, including presented in this manuscript, that being arterial endothelium is a pre-requisite to HSC potential; however, the statement, “we show that the arterial state is a necessary prerequisite for HE formation,” based solely on the fact that *dll4* KD represses arterial and HSC fate, is an overstatement. We note that it is—in highly artificial circumstances—possible to achieve expression of HE/HSC precursor markers in the absence of arterial markers, for example, in Figs. 5 and 6 of Ren, Gomez, Zhang, and Lin, *Blood*, 2010.

We thank the Reviewer for this comment. We specified our conclusion by referring to the DA “*in vivo*” to distinguish it from the manipulated situation of an experiment. We agree that in certain artificial conditions it is indeed possible to induce expression of haematopoietic markers. For example, Burns et al (2005) have shown that heat-shock induced overexpression of the Notch Intracellular domain (NICD) led to ectopic expression of *cmyb* and *runx1* in the vein. However, there is no evidence to demonstrate that such vein cells ectopically expressing *runx1* and *cmyb* will indeed become HSC-like blood progenitors. In addition, there is increasing evidence from recent literature that the arterial identity is indeed critical to make HSCs (see for example Blase and Zon, 2018, *Blood*).

5) There are curious and conspicuous omissions in the references. For example, the authors reference Kissa & Herbomel, 2010 and Boisset et al., 2010 in support of the statement, “During embryogenesis, HSCs are generated from an endothelial precursor termed haemogenic endothelium

(HE),” but not Bertrand et al., 2010, which appeared as trio with the other two articles in the same issue of Nature?

The authors cite Ciau-Uitz & Patient, 2000, North et al., 1999, and Gering and Patient 2005 in support of the statement, “The transcription factor Runx1 is expressed in HE and is essential for the emergence of HSCs,” but not Kalev-Zylinska et al., 2002 and Chen et al., 2009, which are arguably the seminal functional studies on requirement for Runx1 in zebrafish and mouse respectively.

As the authors’ expertise in this field is beyond question, we would suggest reinspecting the referencing of claims to assure that statements are referenced either to reviews, or—if to seminal works—then to the complete sets of seminal works.

We thank the Reviewer for pointing this out and have corrected these omissions.

Reviewer #2 (Remarks to the Author):

The manuscript entitled " Blood stem cell forming haemogenic endothelium derives from arterial endothelium" by Bonkhofer et al utilizes the zebrafish model to examine lineage relationships between the endothelial populations in the embryonic dorsal aorta. The authors develop a novel Runx1 reporter that highlights the earliest onset of Runx1 expression in the aorta in a fairly specific fashion, and go on to characterize cells distinguished in that line in combination with a vascular reporter in both an unbiased and targeted fashion. Large scale profiling analyses indicate that the Runx1+ hemogenic endothelial population is distinct from that of the roof of the aorta, as well as the non-aortic endothelium. This assay also reveals a number of previously uncharacterized factors that seem to be associated with hemogenic fate, including metabolic, inflammatory and transcriptional regulatory factors. Analysis of gradations of reporter gene expression show that the higher the level of

Runx1, the lower the level of arterial associated markers. However, Runx1 reporter expression is consistently associated and indeed appears to be dependent on expression of the arterial marker *dll4*. The authors conclude that HE cells transit through an arterial intermediate, and then down regulate this program via Runx1-mediated repression, however, they do not investigate the necessity of expression of other arterial markers nor the impact of *dll4* over-expression in a population that wasn't normally hemogenic, and thus cannot completely distinguish if there is simply a requirement for Notch signaling in HE, which is also known to essential to arterial commitment, vs the need for an intermediate state. Despite that caveat, this is a comprehensive study of the transcriptional landscape of each of these associated populations, in vivo, including how they are being modified over time, which is of significant interest to the field.

We are pleased that the Reviewer feels that this is a comprehensive study of the endothelial populations in the embryonic dorsal aorta in vivo, which is of significant interest to the field.

Major issues:

1) The authors make an intriguing case for transit through a *dll4*+ arterial intermediate during the commitment to HE fate. However, both *ephrinB2* and *dll4* are Notch targets which had been previously show to function upstream of *gata2* to control Runx1 expression. Are there any non-Notch regulated arterial genes that could be used to confirm this isn't just coincident necessity of Notch for two separate populations rather than a transition through an arterial intermediate?

We thank the Reviewer for this thoughtful comment. By using a *cldn5b*-MO we have been able to show a loss of *runx1* expression, suggesting that it is not simply the Notch input but rather an arterial identity that is required. Strikingly, such *cldn5b* morphants also revealed up-regulation of *dll4*, in line with the relationship between *runx1* and *dll4* that we have identified in this manuscript.

2) A related question arises regarding *dll4* itself. The authors show that *dll4* is needed to express *runx1* in the hemogenic population, however, it would be interesting to know if it is just the Notch signaling aspect that is required, or if you truly have to be an artery. One way to test this would be enforced expression of *dll4* in a population that wasn't normally hemogenic (or one that doesn't become hemogenic until a later time point).

We thank the Reviewer for this comment. This is an intriguing point to discuss. We would like to refer to the study by Burns et al 2005, in which heat-shock induced overexpression of the Notch Intracellular domain (NICD) led to ectopic expression of *cmymb* and *runx1* in the vein. However, there is no evidence to

demonstrate that such vein cells ectopically expressing *runx1* and *cmyb* will indeed become HSC-like blood progenitors. This will require careful lineage tracing experiments and the development of specific tools to address the question. We believe that addressing this question would be a very interesting, independent study that exceeds the scope of this manuscript.

3) The authors do a very nice characterization of expression in the low/med/hi *runx1* population at 29hpf. Since their marker comes on earlier, and since they think HE transitions through an arterial state, if they did the same expression analysis, would they find more arterial markers in the highest *runx1* population, suggesting this is a self-reinforcing continuum?

We thank the Reviewer for this comment. We would like to point to the *dll4* expression data in Fig. 5b and the *efnb2a* expression data in Fig. 6a which also show expression levels at 24/25 hpf. In both cases the expression of such arterial marker genes is higher at the earlier time-point in the DP-R1hi population. Similarly ISH for *dll4* shows stronger expression in the floor of the DA at 24 hpf compared to 27 hpf (Supplementary Fig. 7a).

Minor issues:

4) While the transcriptional associations are interesting, it is unclear how the low/med/hi populations were segregated in the FACS plots due to the thickness of the gating boxes. Are they discrete populations? Or was this an arbitrary fractionation of a continuum.

The DP cells did not form discrete populations in the FACS analysis, which might partly be due to the low cell numbers detected (~18 DP-R1hi cells per embryo). Consequently, when we designed our gating strategy, we used expression of haemogenic marker genes to select for the population with the best enrichment for those genes. This was true when we divided the complete DP fraction into three gates of similar height. We have now added this information to the methods section.

5) Would forced *sox17* expression block HE commitment and/or arterial program down regulation even in the presence of wt *Runx1*?

We thank the Reviewer for this interesting question. We believe that such experiments are outside the scope of our manuscript. Previously published work showed that overexpression of *Sox17* in haemogenic endothelial-like (HE-like) hESCs led to a block in further differentiation of these cells. In addition, *Sox17* overexpression transformed haematopoietic progenitor cells back into HE-like cells (Nakajima-Takagi et al. 2013, Blood), suggesting that indeed forced expression of *Sox17* could be enough to block HE differentiation.

Reviewer #3 (Remarks to the Author):

In these studies, the authors create a Runx1 zebrafish reporter line that can presumably be used to identify and isolate hemogenic and arterial endothelial cells, which they say generate hemogenic endothelial cells. They also use this line to identify Runx1 target genes that may be involved in endothelial to hematopoietic transition (EHT). There are several significant concerns with the studies, as outlined below.

1. There are sites in the developing mouse where hemogenic endothelial cells are specified from primordial endothelial cells, and this process does not involve arterial specification (i.e. yolk sac, placenta). Therefore, the suppression of arterial phenotype is likely specific to the AGM and possibly not relevant to cells in culture, which lack spatial context and microenvironment. This should be considered and discussed with regard to the generation of hemogenic endothelial cells from stem cells in culture, and HSC therefrom.

We thank the Reviewer for this comment. We indeed believe that the arterial identity is a crucial factor that discriminates between HE in the AGM, generating HSCs and the HE in the yolk sac, which cannot generate HSCs. It is tempting to speculate that this is one of the reasons why cell culture differentiation protocols have not achieved the generation of transplantable HSCs without genetic manipulation (defined as the enforced expression of sets of transcription factors) since they miss the arterial specification step before inducing HE. We have now added the following proposition to our discussion: “We propose that it will be crucial to first identify conditions that specifically generate endothelial cells of the aortic lineage of the trunk region before inducing the differentiation into HE-like cells in vitro..”

2. In Figure 1, the expression of Citrine is more intense and more broadly distributed than endogenous Runx1. This is explained as higher protein stability of the reporter, but what is the effect of Runx1 expression in aortic endothelial cells and intersomitic vessels that do not generate HSC?

We thank the Reviewer for this comment. The transgenic cassette only codes for the Citrine protein and not a Citrine-Runx1 fusion protein. Consequently there is no overexpression or ectopic expression of *runx1* in our reporter line. We have now clarified this in the results section and in the schematic. Please see now also Supplementary Fig 1e. Furthermore, we never detect endogenous *runx1* in the roof of the DA or in ISVs by standard ISH.

Interestingly, in our *runx1* overexpression experiment in Supplementary Fig. 8b *cmv* expression was slightly increased in the HE, but no ectopic expression of *cmv* could be detected in the roof of the DA, ISVs or other regions of the embryo. This further indicates that *runx1* functions on cells that already have haemogenic potential.

3. In a related issue, Runx1 is not a “master” inducer of hemogenic endothelial cells (nor a specific marker) in zebrafish (as also determined in mouse by several groups) or the other endothelial cells that express it would generate HSC. This important issue should be addressed, especially since hemogenic endothelial cells are defined as Runx1+, and sorted based on Runx1 expression. Clearly, other endothelial cells express Runx1, and Runx1 expression does not induce a hemogenic phenotype.

We agree with the Reviewer that Runx1 does not induce HE, which we now mention in our discussion, but is needed for further differentiation and initiation of EHT.

We also agree that the definition of HE in zebrafish is not as elaborate as in the mouse system. Therefore, we already felt the urge to clearly state that the definition of the HE as *kdrl+* *runx1+* (DP) endothelium, which is an accepted one in the zebrafish model (*runx1* expression in the DA is widely used as a bona fide HE marker) is a working definition, which we now clearly state. In this regard we never detected mCherry+ Citrine+ cells outside of the DA by fluorescence microscopy at the timepoints of our analysis (24-34 hpf). Similarly, in the mouse *Runx1* expression in endothelium is restricted to sites of blood stem and progenitor cell emergence (North 1999 and others) and is generally considered a marker of HE.

4. There is no evidence that *Runx1+* endothelial cells give rise to *Runx1-* endothelial cells, so this suggestion should be excluded from the Results sections.

We agree with the Reviewer that there is no evidence that endothelial cells with a stable expression of *runx1* will become *runx1*^{negative}. During the very dynamic process in which DA angioblasts migrate to the midline and form a cord that then starts to lumenise, we detect initial expression of *runx1P2:Citrine* in most DA cells. However, that expression only becomes stable in the cells that remain in the DA floor. Only those cells express levels of *runx1* that are detectable by ISH.

5. These studies suggest that *Runx1* regulates Notch signaling, but does Notch signaling directly regulate *Runx1* expression in the zebrafish aorta? In mouse yolk sac and aortic hemogenic endothelial cells, *Runx1* is upregulated downstream of Notch signaling. In addition, this occurs downstream of retinoic acid signaling and cKit expression, so what is the temporal status of cKit expression and retinoic acid signaling, relative to Notch activation, during zebrafish aorta definitive hematopoiesis?

We thank the Reviewer for these questions. Similar to the situation in mouse, *runx1* is also downstream of Notch signaling as initially shown by Burns et al (2005) and different Notch inputs might be required at multiple stages. It was shown that early somite-derived Notch signals through the ligands *Dlc* and *Dld* (Clements et al., 2011) and establish HSC fate/competence as vascular precursors migrate across the ventral face of the somites (Kobayashi et al., 2014). It was also shown that the ligand *Jag1a* is required for haemogenic expression of *runx1* (Butko et al., 2015, Monteiro et al, 2016). A recent publication suggests that *kitb*/*kitlgb* are enriched in the EMP and HSC populations. They also show that they are upstream regulators of *runx1* expression in HE between 22-36hpf (Mahony et al, 2018, Stem Cell Reports), a developmental window where Notch signalling is also required. However, the link between *kit* and RA or Notch signalling has not been explored.

6. There is no direct evidence presented to show that *Runx1* represses arterial gene expression, and this is a main conclusion.

We agree with the Reviewer that we do not show any evidence of direct repression of arterial genes by *runx1*. However, we do see the upregulation of arterial genes in *runx1* mutants which is likely indirect through upregulation of transcriptional repressors including *Gfi1* or downregulation of transcriptional

activators including Sox17. We have now modified the text in the discussion to indicate that the question of direct or indirect repression has not yet been resolved.

7. The use of morpholinos to suppress gene expression, especially without MO + gene rescue controls, is no longer considered an acceptable approach due to potential off-target effects.

We agree with the Reviewer that the use of MOs has to be tightly controlled to avoid misinterpretation. In the case of the *runx1* MO, it had already been shown to mimic the *runx1* mutant (Sood, et al. 2010 Blood 115, 2806–2809). Additionally, we designed a novel strategy to control for off-target effects in our RNAseq experiments, where we analyzed for differentially expressed genes in a *runx1*-negative cell population. Such genes will most likely all be off-targets, so we eliminated them from our potential target list. In this regard, *tp53* is commonly upregulated in MO experiments and was identified as an off-target. In agreement with this, *tp53* was not differentially expressed in *runx1* mutants (see below). Furthermore, all potential target genes identified by our MO-based RNAseq experiment were validated by qPCR in *runx1* mutants.

qRT-PCR gene expression analysis of potential Runx1 (off-)targets (*tp53*) in the HE and ARE of *runx1*^{+/+} (WT) and the HE of *runx1*^{-/-} mutant (MUT) embryos. *tp53* was identified as an MO-off target by our RNAseq analysis. qPCR analysis in MUT verified *tp53* as a true *runx1*-MO off target.

In the case of the *dll4* morpholinos, we observed the same phenotype (loss of *runx1* expression in the HE) with two different published morpholinos; in addition, Leslie et al, 2007 (Development) had shown previously that there was concordance between *dll4* MO and mutant phenotypes. Importantly, we have now incorporated *dll4* mutant data (Fig 6f).

8. In the ATAC analysis, the “hemogenic endothelial cells” are likely enriched for open sites related to Runx1 regulation because the analyzed cells were sorted based on Runx1 expression. Also, there are no mechanistic studies to demonstrate the functionality of putative Runx1 targets that may play a role in EHT.

We thank the Reviewer for this comment. The ATAC-seq data was mainly used to further validate our BAC-based *runx1*-reporter line and our gating strategy. With respect to the Runx1 target genes, it remains to be investigated whether the identified genes play a role during EHT or if they are involved in later processes. We feel that such analyses are beyond the scope of this manuscript and will be addressed in subsequent studies. However, one of the *runx1* target genes we identified, *dnmt3bb.1* (see Supplementary fig 6b), is required during definitive haematopoiesis (Gore et al, 2016). We have added this to the manuscript. Thus, the identification of Runx1 target genes is a validation of our gating

strategy to successfully sort cells of the HE and the DA, which subsequently allowed us to make statements about the expression of arterial and venous genes within these populations.

Reviewers' Comments:

Reviewer #1:

Remarks to the Author:

SUMMARY AND RECOMMENDATION

Bonkhofer et al. present a revised manuscript supporting the idea that hemogenic endothelium derives from an arterial endothelial precursor, and that initiation of the definitive hematopoietic program through a second phase expression of *runx1* involves a binary choice between arterial endothelium and definitive hematopoietic precursor with HSC capability. We continue to find the manuscript well written, compelling, and of high interest on an important topic. Many of our concerns were addressed in the revision. We continue to have some minor concerns, that we feel it would be useful to see addressed in a published version.

MINOR

1. The *kdrl*-SP population is used as the control comparator for "non-arterial endothelium" (NAE), which is depicted as uniquely venous endothelium of the cardinal vein in Fig. 2e of the current manuscript. The *kdrl*-SP population is sorted from 28-30 hpf double transgenic embryos to remove SP-*runx1* (neurons?) and double positive cell populations. The authors also treat it as venous endothelium for the purpose of defining the comparative identities of their profiling data sets. It is therefore critical that this population represent venous endothelium. We previously pointed out that another putative endothelial specific transgenic reporter (*fli1a*:EGFP) does not perfectly overlap with *kdrl* reporter transgenics. For example, if double transgenic *kdrl*:mCherry;*fli1a*:EGFP populations are sorted, there are obvious single positive populations for both reporters. This result indicates that either there are unique *fli1a*⁺ and *kdrl*⁺ varieties of endothelium or that (more likely) either or both transgenic animals label non-endothelial tissues that would contaminate a single positive population such as *kdrl*-SP. In support of the idea that the *kdrl*-SP cells are venous, the cells do express low, but significant levels of *nr2f2* (a.k.a *coup-tfii*; Fig. S4 c). However there are several results that decrease confidence that *Kdrl*-SP cells are in fact mostly or completely venous:

First, the *kdrl*-SP population express the highest levels of the arterial markers *efnb2a* (Fig. 2d) and *dlc* (Fig. S4). They also express the arterial roof endothelium marker *tbx20* at levels far higher than any of the other sorted and profiled populations (Fig. 2d), which is especially surprising and unremarked, given that the authors themselves acknowledge it as a "DA-roof marker" (page 8). Second, they express low but apparently significant levels of the myotome marker *myod1* and the sclerotome/neural crest marker *pdgfra*. Third, the "cluster" of upregulated genes that is unique to putatively venous *kdrl*-SP cells (cluster 7, presented but not discussed in the text) does not contain obviously venous markers, but rather mostly cell proliferation markers (*hmmr*, *mki67*, *nusap*, and cyclin related genes—*cdc20*, *cdca8*, *ccnb1*), CNS markers (*gnao1b*, *hirip3*, *wrb*, *aspm*, and *mibp*) and one obvious muscle pioneer cell marker, *mustn1a*. We therefore continue to have concerns about the identity of the cells attributed to this gene set.

2. The authors have included *pdgfra* and *cdh17* as requested but misidentified the tissues they label: *pdgfra* labels sclerotome/neural crest and *cdh17* labels pan-pronephros.

3. The cluster analysis overall is confusing. It is presented as evidence of the correct assignment of identities for the various populations (e.g. "Consensus clustering analysis (Supplementary Fig. 5a-c) verified the endothelial character of DP-R1lo cells that shared a substantial gene set with endothelial cells of the SP-*kdrl* gate (Cluster 1, 2570) genes." Manuscript p. 10). And it is true that both venous (*dab2*—apparently not uniquely venous as sometimes presented, *flt4*, *mrc*, *stab2*, *ephb4a*, *ftr82*, *atp1b1a*), arterial (*dlc*, *notch1b*, *efnb2a*), and pan-endothelial (*cdh5*, *kdrl*, *etv2*) markers appear in Cluster 1; however, so also do somite (*meox1*, *myh9a*), sclerotome (*pax1a*, *pax9*), neural crest (*sdc4*, *aox5*), sclerotome/neural crest (*pdgfra*, *twist1b*, *twist2*, *snai2*), heart (*myl7* a.k.a *cmlc2*), hypochord (*col2a1a*), floorplate (*shha*), and pronephric markers (*cdh17*, *ret*, *pax2a*, *atp1b1a*) in a cursory analysis. Many of these non-endothelial markers are routinely

considered to be black-and-white tissue specific marker genes, for example myl7 and cdh17. It is therefore unclear why the authors have concluded that this cluster analysis “verified the endothelial character” of the sort populations.

We have similar concerns about other clusters.

4. The utterly convincing results demonstrating that sox17 is repressed by Runx1, which the authors interpret to mean that Sox17 helps form hemogenic endothelium but must be turned off to form HSC precursors appear to conflict with the findings of Kim, Saunders, and Morrison (Cell, 2007), which identified Sox17 as a sort of fetal hematopoietic stem cell maintenance factor. Could the authors address this conflict or explain why it is not a conflict?

5. The angpt1 results presented in Fig. 4h are difficult to see and interpret.

6. We believe that given the absence of functional validation for the attribution of novel Runx1 targets, the authors might consider softening from “we have identified a diverse set of target genes” to “potential target genes.”

Reviewer #2:

Remarks to the Author:

My concerns have been addressed.

Reviewer #3:

Remarks to the Author:

The additions and clarifications to the manuscript have improved the story overall. However, given that only hemogenic endothelium in the “zebrafish aorta” was examined, the title should be modified to clarify that.

MINOR

1.

The *kdrl*-SP population is used as the control comparator for “non-arterial endothelium” (NAE), which is depicted as **uniquely venous endothelium** of the cardinal vein in Fig. 2e of the current manuscript. The *kdrl*-SP population is sorted from 28-30 hpf double transgenic embryos to remove SP-*runx1* (neurons?) and double positive cell populations. **The authors also treat it as venous endothelium for the purpose of defining the comparative identities of their profiling data sets.** It is therefore critical that this population represent venous endothelium. We previously pointed out that another putative endothelial specific transgenic reporter (*fli1a*:EGFP) does not perfectly overlap with *kdrl* reporter transgenics. For example, if double transgenic *kdrl*:mCherry;*fli1a*:EGFP populations are sorted, there are obvious single positive populations for both reporters. This result indicates that either there are unique *fli1a*⁺ and *kdrl*⁺ varieties of endothelium or that (more likely) either or both transgenic animals label non-endothelial tissues that would contaminate a single positive population such as *kdrl*-SP. In support of the idea that the *kdrl*-SP cells are venous, the cells do express low, but significant levels of *nr2f2* (a.k.a *coup-tfii*; Fig. S4 c). However there are several results that decrease confidence that *Kdrl*-SP cells are in fact mostly or completely venous:

First, the *kdrl*-SP population express the highest levels of the arterial markers *efnb2a* (Fig. 2d) and *dlc* (Fig. S4). They also express the arterial roof endothelium marker *tbx20* at levels far higher than any of the other sorted and profiled populations (Fig. 2d), which is especially surprising and unremarked, given that the authors themselves acknowledge it as a “DA-roof marker” (page 8). Second, they express low but apparently significant levels of the myotome marker *myod1* and the sclerotome/neural crest marker *pdgfra*. Third, the “cluster” of upregulated genes that is unique to putatively venous *kdrl*-SP cells (**cluster 7**, presented but not discussed in the text) does not contain obviously venous markers, but rather mostly cell proliferation markers (*hmmr*, *mki67*, *nusap*, and cyclin related genes—*cdc20*, *cdca8*, *ccnb1*), CNS markers (*gnao1b*, *hirip3*, *wrb*, *aspm*, and *mibp*) and one obvious muscle pioneer cell marker, *mustn1a*. We therefore continue to have concerns about the identity of the cells attributed to this gene set.

Due to the similar nature of question #3 with the second part of question #1 we would like to respond to them together.

3.

The cluster analysis overall is confusing. It is presented as evidence of the correct assignment of identities for the various populations (e.g. “Consensus clustering analysis (Supplementary Fig. 5a-c) **verified the endothelial character** of DP-R11o cells that shared a substantial gene set with endothelial cells of the SP-*kdrl* gate (Cluster 1, 2570) genes.” Manuscript p. 10). And it is true that both venous (*dab2*—apparently not uniquely venous as sometimes presented, *flt4*, *mrc*, *stab2*, *ephb4a*, *ftr82*, *atp1b1a*), arterial (*dlc*, *notch1b*, *efnb2a*), and pan-endothelial (*cdh5*, *kdrl*, *etv2*) markers appear in **Cluster 1**; however, so also do somite (*meox1*, *myh9a*), sclerotome (*pax1a*, *pax9*), neural crest (*sdc4*, *aox5*), sclerotome/neural crest (*pdgfra*, *twist1b*, *twist2*, *snai2*), heart (*myl7* a.k.a *cmlc2*), hypochord (*col2a1a*), floorplate (*shha*), and pronephric markers (*cdh17*, *ret*, *pax2a*, *atp1b1a*) in a cursory analysis. Many of these non-endothelial markers are routinely considered to be black-and-white tissue specific marker genes, for example *myl7* and *cdh17*. It is therefore unclear why the authors have concluded that this cluster analysis “verified the endothelial character” of the sort populations.

We have similar concerns about other clusters.

We thank the reviewer for these comments, for the thorough investigation of our dataset and for sharing their thoughts on *kdrl*⁺ endothelial cells in relation to *fli1a*⁺ cells. In the following we respond to the individual aspects raised in comment #1 and #3.

- We believe that most of the concerns raised can be answered by resolving a misunderstanding. The abbreviation NAE stands for non-**aortic** endothelium rather than **non-arterial** endothelium, as introduced at the beginning of the second paragraph in the results section: “...but not in non-aortic endothelium (NAE) including the cardinal vein (CV) and veins and arteries in the head and tail” Consequently we expected to see expression of arterial markers like *efnb2a* or *dlc* in the NAE population, explaining their presence in Cluster-1. However crucial arterial marker genes like *dll4* and *hey2* had the highest expression in the DP-R1¹⁰ population.

→ To avoid misinterpretation of the NAE term we added additional clarification in the text when introducing our gating strategy: “Cells single-positive for *kdrl:mCherry+* (SP-*kdrl*), **representing a mixture of venous and arterial cells**, served as a NAE control”.

- Since our cluster analysis compared different endothelial tissues to each other, we expected to see the vast majority of the endothelial programme shared and not unique to one specific cell type. **Cluster-7** contains genes with higher expression only in the SP-*kdrl* population and indeed the important venous marker gene *nr2f2* can be found in Cluster-7 (see Fig. S4c). However, SP-*kdrl* cells (representing NAE) are not a pure fraction of venous cells but a mixture with arterial cells. Consequently the venous programme within this population is diluted. Also, at this time of development, even though venous and arterial cell types are already separated, expression of certain marker genes are not yet exclusive to either tissue type. For example the venous marker genes *flt4* and *ephb4a* are not exclusively venous at the timepoint of our analysis (see below images from Thisse et al., 2001 taken from Zfin.org). In both cases expression can still be found in the aortic region as well as the cardinal vein. In contrast, expression of *nr2f2* seems to be restricted to the cardinal vein already around 24hpf (Fig. 5A in Swift et al., 2014; see figure below taken from Zfin.org). Together, these particularities explain why most venous genes are found in Cluster-1.
→ As pointed out in the manuscript, we do find most pan-endothelial marker genes in Cluster-1 (with higher expression in the SP-*kdrl* and DP-*RI^{lo}*), indicating an active endothelial programme in DP-*RI^{lo}* cells, which we think is crucial information to provide. With respect to the concerns raised, we changed the wording in the manuscript from “verified the endothelial character” to “Consensus clustering analysis **further supported** the endothelial character of DP-*RI^{lo}* cells”.
Lastly, the 41 genes of Cluster-7 mostly contained genes attributed to “cell cycle”. We speculate that cells in the SP-*kdrl* population are undergoing more frequent angiogenesis than cells in the DP-*RI* populations.

<Published image redacted: but see <https://zfin.org/ZDB-IMAGE-030429-766> >

Thisse et al. 2001 *ephb4a* (ZFIN)

<Published image redacted but see <https://zfin.org/ZDB-IMAGE-040107-49> >

Thisse et al. 2001 *flt4* (ZFIN)

<Published image redacted - see <https://zfin.org/ZDB-IMAGE-031002-260> >

Swift et al., 2014 Fig.5a (ZFIN), 24hpf

- *Tbx20* (and *kdrl*) is also expressed in the endothelial lining of the heart (endocardium) during embryogenesis (see below *tbx20* expression from Thisse et al, 2001 in Zfin.org). This endothelium most likely is included within the NAE population, thus explaining the presence of *tbx20* in the SP-*kdrl* population.
In the context of all DP-R1 cell populations, expression levels of *tbx20* distinguish DP-R1^{lo} cells (DA-roof) from DP-R1^{hi} cells (DA-floor/HE).

<Published image redacted>

Thisse et al. 2001 *tbx20* (ZFIN)

- We agree with the reviewer that the information regarding the non-perfect overlap between *kdrl*⁺ and *fli1a*⁺ reporter lines, both of which are thought to mark endothelium, indeed indicates that additional cell types might be marked to a certain degree. This is a common drawback of fluorescent reporter lines and one that cannot be resolved. While we cannot rule out contamination by non-endothelial cell types within our sorting strategy based on the *kdrl:mCherry* and *runx1P2:Citrine* reporter lines, we believe those contaminations are minimal, exemplified by the data provided in the supplementary information (Sup Fig.4c) and addressed in the manuscript.
For the clustering analysis we first performed an ANOVA across the 4 “endothelial” populations, which identified 5574 differentially expressed genes (FDR < 0.05). We excluded 361 genes with a log2-FC < +/- 0.7. Since the number of the remaining 5213 genes was appropriate for the clustering analysis, we did not perform further filtering to exclude low expressed genes. Furthermore, we performed bulk sequencing. Therefore we cannot resolve to what extent the genes mentioned by the reviewer are detected due to contamination by non-endothelial cell types or due to low expression within certain endothelial cells. With respect to the two mentioned “black-and-white” tissue markers *myl7* and *cdh17*, these genes have very low TPM values as shown in the table below. Note that *myl7* expression was found in the SP-*kdrl* population because it is also expressed in the endocardium and as explained above, would therefore be included in the SP-*kdrl* population).
→Importantly, minimal cell contamination seems to mostly concern the SP-*kdrl* population. We highlighted that in the manuscript by modifying the existing text as below: “As expected, *nr2f2* expression was only enriched in the SP-*kdrl* population (Supplementary Fig. 4c). The low TPM values for *myod1*, *pdgfra* and *cdh17* suggest only minimal contamination by cells of the myotome, sclerotome/neural crest or pronephros, respectively, **mostly within the SP-*kdrl* population.**”
Since we later compare the DP-R1^{hi} with the DP-R1^{lo} population to investigate the HE in relation to the ARE, we do not believe that these particularities compromise our further analysis

Table: Average TPM value within the respective cell populations

Genes	SP- kdrl	DP-R1-high	DP-R1-medium	DP-R1-low
-------	-----------------	------------	--------------	-----------

Heart				
myl7	17.3	0.0	0.1	0.8
Pronephric				
cdh17	2.7	0.0	0.0	5.2

2.

The authors have included pdgfra and cdh17 as requested but misidentified the tissues they label: pdgfra labels sclerotome/neural crest and cdh17 labels pan-pronephros.

We thank the reviewer for pointing out this mistake. We will correct it in the paper.

4.

The utterly convincing results demonstrating that sox17 is repressed by Runx1, which the authors interpret to mean that Sox17 helps form hemogenic endothelium but must be turned off to form HSC precursors appear to conflict with the findings of Kim, Saunders, and Morrison (Cell, 2007), which identified Sox17 as a sort of fetal hematopoietic stem cell maintenance factor. Could the authors address this conflict or explain why it is not a conflict?

The excellent work by Kim, Saunders, and Morrison (Cell, 2007) shows a thorough analysis of the role of Sox17 in the generation of fetal and neonatal HSCs in contrast to adult HSCs. The fact that Sox17^{GFP/GFP} mice did not develop visible haematopoiesis at E11.5 implies a crucial earlier function, also indicated by the transplantation experiments performed around E12.5 (also with conditional Tie2-Cre mice). However within their work, they do not address the timing of Sox17 expression before E14.5 (including early on during HE specification).

In mouse embryos, expression of Sox17 can be detected in the HE (Choi et al., 2012; Lizama et al., 2015). Furthermore, an inverse relationship between Sox17 and Runx1 expression levels in the HE has been described (Bos et al., 2015; Chen et al., 2016; Lizama et al., 2015). Sox17 is required for priming haemogenic potential and “locks” the HE state (Clarke et al., 2013; Nakajima-Takagi et al., 2013). Furthermore, overexpression of Sox17 reprogrammed haematopoietic progenitors into HE-like cells (Nakajima-Takagi et al., 2013). Direct repression of Sox17 by Runx1 was also suggested previously *in vitro* during differentiation culture of early blood progenitors (EBPs) or in human umbilical artery endothelial cell (HUAEC) culture (Lizama et al., 2015; Tanaka et al., 2012). Here we provide the first functional *in vivo* data indicating that Runx1 is indeed repressing sox17 expression within the aortic HE.

We speculate that Sox17 is re-expressed after HSC emergence and migration to the FL (CHT/kidney in zebrafish) where it is required to maintain fetal and neonatal HSCs, as shown by Kim, Saunders, and Morrison (Cell, 2007). Strikingly, similarly to the reprogramming of haematopoietic progenitors into HE-like cells by Sox17 overexpression (Nakajima-Takagi et al., 2013), a later study by the Morrison lab (He *et al.* 2011) showed that ectopic expression of Sox17 in adult HSCs was sufficient to confer fetal HSC characteristics on adult HSCs (increased self-renewal potential and expression of fetal HSC genes). In this light, Sox17 might have very similar functions during two different steps in the HSC differentiation process: during HSC *de novo* generation from HE, and during the expansion and final transition phase from foetal to adult HSCs. In both cases the presence of Sox17 keeps the respective cell types “locked” and for subsequent differentiation the repression/down-regulation of Sox17 is required.

5.

The angpt1 results presented in Fig. 4h are difficult to see and interpret.

We thank the reviewer for pointing out the requirements for a zoom of the crucial area, which we will provide.

6.

We believe that given the absence of functional validation for the attribution of novel Runx1 targets, the authors might consider softening from “we have identified a diverse set of target genes” to “potential target genes.”

We thank the reviewer for this comment. We will rephrase our conclusion to “we have identified a diverse set of potential target genes”.

--

Reviewer #2 (Remarks to the Author):

My concerns have been addressed.

--

Reviewer #3 (Remarks to the Author):

The additions and clarifications to the manuscript have improved the story overall. However, given that only hemogenic endothelium in the "zebrafish aorta" was examined, the title should be modified to clarify that.

We modified the title accordingly:

“Blood stem cell-forming haemogenic endothelium in zebrafish derives from arterial endothelium”